# Piezo-generated charge mapping revealed through direct piezoelectric force microscopy

A. Gomez[1], M. Gich[1], A. Carretero-Genevrier [ID] [2], T. Puig[1] & X. Obradors[1]

While piezoelectric and ferroelectric materials play a key role in many everyday applications, there are still a number of open questions related to their physics. To enhance our understanding of piezoelectrics and ferroelectrics, nanoscale characterization is essential. Here, we develop an atomic force microscopy based mode that obtains a direct quantitative analysis of the piezoelectric coefficient $d_{33}$. We report nanoscale images of piezogenerated charge in a thick single crystal of periodically poled lithium niobate (PPLN), a bismuth ferrite ($BiFO_3$) thin film, and lead zirconate titanate (PZT) by applying a force and recording the current produced by these materials. The quantification of $d_{33}$ coefficients for PPLN ($14 \pm 3$ pC per N) and BFO ($43 \pm 6$ pC per N) is in agreement with the values reported in the literature. Even stronger evidence of the reliability of the method is provided by an equally accurate measurement of the significantly larger $d_{33}$ of PZT.

[1] Institut de Ciència de Materials de Barcelona (ICMAB-CSIC), Campus UAB, Bellaterra, Catalonia 08193, Spain. [2] Institut d'Electronique et des Systemes (IES), CNRS, Universite Montpellier 2 860 Rue de Saint Priest, Montpellier 34095, France. Correspondence and requests for materials should be addressed to A.G. (email: agomez@icmab.es)

The piezoelectric effect, which consists in the dielectric polarization of non-centrosymmetric crystals under a mechanical stress, was discovered by the Curie brothers in 1880[1]. The following year, from thermodynamic considerations, G. Lippmann predicted the converse effect, i.e., that a piezoelectric material would be mechanically strained by an applied electric field[2] and the Curies readily measured it[3]. These findings spawned more research which eventually led to the discovery of ferroelectricity in polar piezoelectrics[4]. Since those early discoveries, the unique ability of piezoelectrics and ferroelectrics for interconverting mechanical and electrostatic energies[5] has endlessly inspired technological developments and these materials, which represent nowadays a billion euro industry, are found in many everyday applications[6–12]: ultrasound generators for echography scanners, shock detectors within airbags, accelerometers, diesel injection valves, tire pressure sensors, vibration dampers, oscillators, improved capacitors, or new dynamic access random memories, to just cite a few. Moreover, the prospects for future applications in new markets are bright, including energy harvesting, CMOS replacement switches, or photovoltaics and photocatalysis[13–16]. Yet, in spite of such industrial relevance and the amount of past and present research, the basic understanding of piezoelectricity and ferroelectricity is challenged and reshaped by findings that come along with new developments in the material characterization. This is well illustrated by the advances in Atomic Force Microscopy (AFM), which brought a new perspective of ferroelectric domain walls[17–20]. The development of new modes with improved spatial resolution have revealed the domain wall complexity and its intrinsic properties[21–23] and have also opened the door to get more insight in long-date issues such as the extrinsic contributions to dielectric permittivity and piezoelectricity due to domain wall pinning at dislocations and grains[24]. In this direction, Piezoresponse Force Microscopy (PFM) is the most widely used technique for the nanoscale and mesoscale characterization of ferroelectric and piezoelectric materials[25–28]. PFM method is based on the converse piezoelectric effect and consists in measuring the material deformation under an AC electric field applied through the contacting AFM tip. In this technique the sample vibration is determined by the tip displacement, which is an indirect measurement[29], making the accurate determination of the piezoelectric coefficient challenging. Moreover, the quantitative piezoelectric measurements by PFM[30], are further complicated by the difficulty of disentangling, from the electromechanical response, the contributions of the piezoelectric response and other physical phenomena such as, ionic motion and charging, electrostatic or thermal effects[18, 31–33]. Indeed, the increasing awareness about these issues among the scientists of the field[34] prompts the need for new developments in scanning probe microscopies, which remain a unique tool for the characterization of piezoelectric and ferroelectric materials at the nanoscale.

It has been previously shown that surface charge scraping can occur if a ferroelectric sample is scanned at high speed with an AFM tip[35–38]. The charge scrapped by this mechanism is proportional to the surface charge density of a ferroelectric material. Such mode, proposed by Hong et al. is named Charge Gradient Microscopy (CGM) and images ferroelectric screening charges[37]. It is important to note that the work of Hong et al. shows that the contribution of piezoelectric charge in CGM experiments is negligible compared to unscreened displacement charge and hence that CGM signal is not related to the piezoelectric properties of the sample. In order to avoid any contribution of CGM scraped charges, we designed our method as follows. The samples were scanned at extremely low speeds, 1000 times slower than in CGM experiments, and under a considerable load, up to 100 times larger than in CGM. We have included a complete comparison of the results of both methods further in this text.

To address this need, here we introduce a SPM tool that exploits the direct piezoelectric effect to obtain a quantitative measurement of the piezoelectric constant in ferroelectric materials. This technique, that we name Direct Piezoelectric Force Microscopy (DPFM) uses a specific amplifier and a conductive tip which simultaneously strains a piezoelectric material and collects the charge built up by the direct piezoelectric effect. We studied

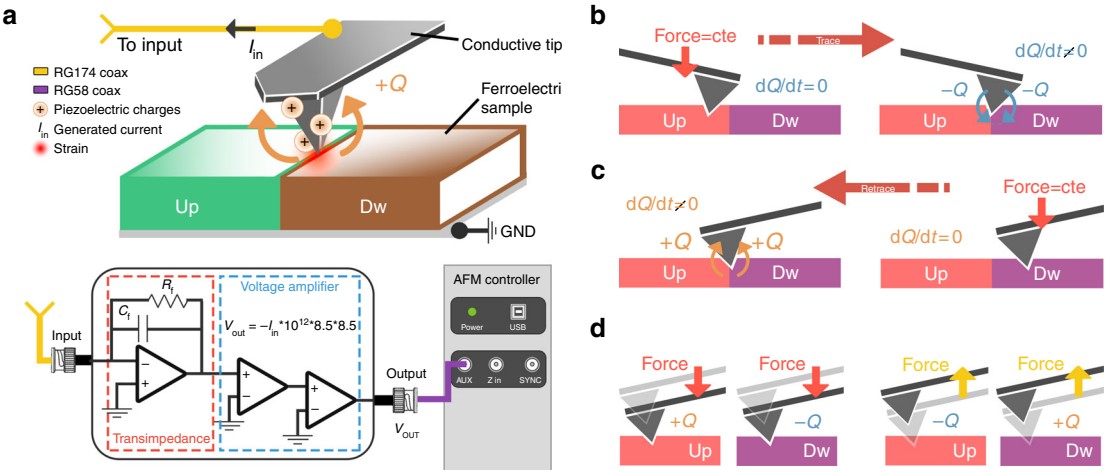

**Fig. 1** Schematic of the measurement set-up and explanation of the proposed experiments. **a** Set-up used to record the piezoelectric charge generated by the material through the use of a special current-to-voltage transimpedance amplifier. The amplifier maintains a reasonable bandwidth of 4–5 Hz with an ultralow input-bias current consumption of <0.1 fA. **b** When a single domain polarization is scanned, there is no current flowing, as the force is kept constant. However, when the tip crosses different domains, there will be a current flowing as the force is kept constant, but the $d_{33}$ value will invert its sign. **c** At the domain wall, the generated current can be modeled as the tip will enter into one domain, loading it, and will leave an opposite domain, unloading it. The sign of charge generation depends on the scan direction or more precisely on whether the tip crosses from and Up to a Down domain (generation of $Q$ < 0) or from an Down to an Up domain (generation of $Q$ > 0). **d** Spectroscopy sweep model obtained when the tip performs a force-vs-distance sweep. While the tip exerts a force on the sample a strain is created. Once the force is released, unstraining occurs. The straining and unstraining processes generate positive or negative charges depending on the polarization of the domain

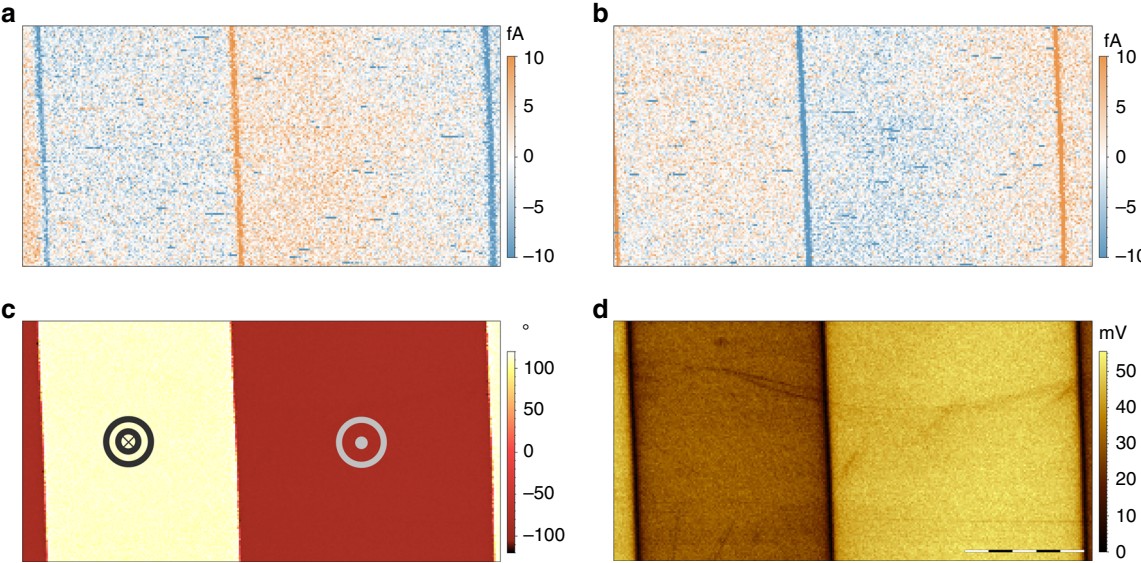

**Fig. 2** Piezogenerated charge of periodically poled lithium niobate (PPLN) **a** Piezo-generated current map of PPLN obtained when the tip scans from left to right-trace (DPFM-Si). **b** Piezo-generated current map of PPLN obtained when the tip scans from right to left-retrace (DPFM-So). Current is generated at domain walls, orange and blue vertical lines, where the tip strains and unstrains domains of opposite polarization. Inside the domains, a near zero current signal is observed, however a small contrast is present that can be due to surface screening recharging process. **c** PFM phase image and **d** PFM amplitude image of the same sample, obtained simultaneously with DPFM signals, scale bar 7.5 µm. In order to obtain DPFM signals an AC bias was connected to the back electrode of the specimen

the feasibility of this new mode by exploring the piezogenerated charges of periodically poled lithium niobate (PPLN), bismuth ferrite (BFO), and lead zirconate titanate (PZT) ferroelectrics.

## Results

**Measuring piezogenerated charges with an AFM.** The amplifier is an ultralow input-bias current (<0.1 fA) transimpedance capable of converting electric current into a voltage signal, readable by any commercial microscope (Fig. 1a). As a consequence, the developed set-up has a very low leakage current, and thus all the charges generated by the piezoelectric material can be read by the amplifier. Just by maintaining the tip on the surface of the films and sequentially applying different force values with the AFM tip, the charges generated by the material are measured and the direct piezoelectric coefficient can be readily calculated from the applied stress and the collected compensation charge. Interestingly, by combining this tool with PFM measurements, a complete electromechanical and piezo-charge generation characterization can be achieved. Measuring the direct piezoelectric effect with an AFM is a challenge that has not been addressed so far due to the impossibility of performing reliable measurements of tiny amounts of generated charge. An AFM probe can apply a user predefined force with picoNewton precision, up to maximum values of hundreds of microNewtons[39–41]. Applied to a piezoelectric material, such force will generate a charge, which can be collected to obtain currents of different intensity depending on the sampling time. For instance, we can estimate that the 1 fC charge generated by applying a 100 µN force into a 10 pC N$^{-1}$ piezoelectric material[42], will produce a current of 1 fA if generated in 1 s, 2 fA if generated in 0.5 s and so on. With such requirements, an amplifier capable of measuring 1 fA with a BandWidth of 1 Hz is needed. More importantly, the charge that the amplifier leaks has to be well below that desired threshold of 1 fA, otherwise a substantial part of the current will be lost during measurements. Since these requirements were not met by any AFM manufacturing companies, a special amplifier was employed.

**Experimental set-up.** The measurements were made with a commercial AFM Keysight 5500 LS. The complete set-up to perform the experiments according to the proposed method is depicted in Fig. 1a. The amplifier consists of three different commercially available operational amplifiers, which were supplied by Analog Devices Inc. The amplification process is divided into two stages, a transimpedance stage and a voltage amplifier stage. The transimpedance stage was configured with a feedback resistor of 1TeraOhm which yields a current-to-voltage gain of $-1 \times 10^{12}$ V A$^{-1}$ [43]. The voltage amplifier stage adds an additional gain of 72.25. Following standard amplifier theory, the final gain of both concatenated stages is the multiplication of each stage gain, which results in a gain of $-72.25 \times 10^{12}$ V A$^{-1}$ [44]. Even though theoretical gain calculation is precise, we experimentally calibrated the amplifier twice with a test resistor of $40 \pm 0.4$ GOhm giving an experimental gain of $-16.9 \pm 1.0 \times 10^{12}$ V A$^{-1}$ (Supplementary Fig. 1). The leakage current through the amplifier induces an error, which will be responsible of charge losses while measuring. Such current was provided by Analog Devices as being as low as 0.1 fA, which can be considered small compared to the generated piezocharge[45] to be measured, which is in the order of several fA. An intrinsic property of the set-up is that both tip and back-surface of the sample are connected to ground, which enables the study of high leakage ferroelectric films.

With such set-up the charge generated by a piezoelectric material can be recorded with an AFM tip. The physics underlying the generated current is depicted in Fig. 1b–d. Two different cases are considered, when the tip scans from left to right (Trace) and from right to left (Retrace). While in trace scanning, Fig. 1b when a single domain polarization is scanned, there is no current flowing, as the force is kept constant. However, when the tip crosses different domains, there will be a current flowing as the force is kept constant, but the $d_{33}$ value will invert its sign. At the domain wall, the generated current can be modeled as the tip will enter into one domain, loading it, and will leave an opposite domain, unloading it. Such strained and unstrained mechanism will indeed generate a current. Similarly,

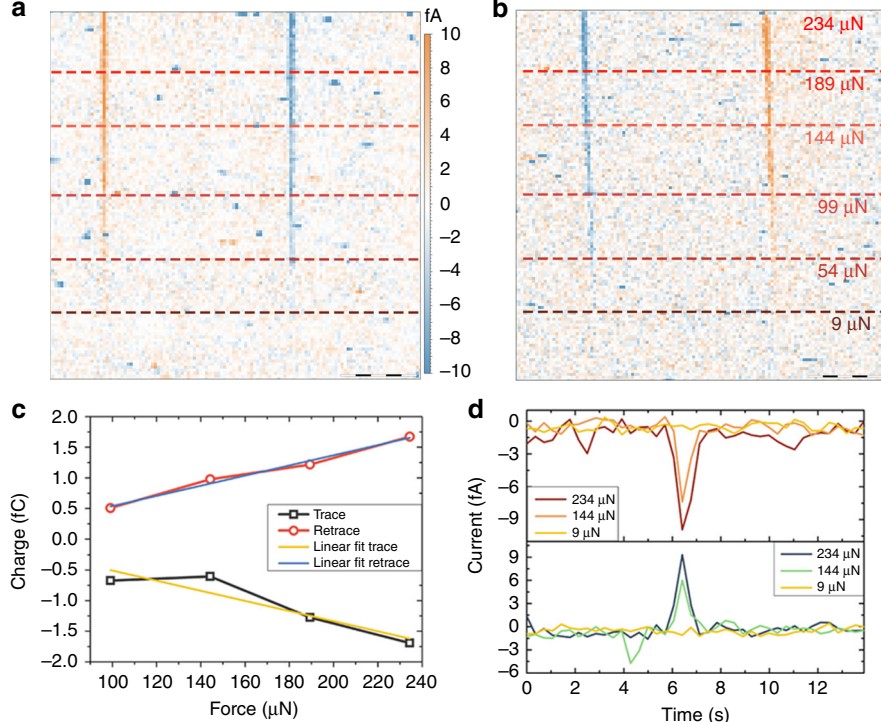

**Fig. 3** Force dependence for the piezogenerated charge mapping. **a** DPFM-Si and **b** DPFM-So of the proposed PPLN test sample, obtained at different applied forces, scale bar 5 μm. In order to demonstrate the origin of the recorded current, different forces where applied during the scan-see red line dot. The current recorded increases with the applied force, as expected from a piezoelectric charge generation. **c** charge vs force spectroscopy sweep obtained from the profiles of Fig. 3a. The current profiles where integrated with a time constant of 390 ms in order to obtain the charge generated at a specific pixel. The linear relation displayed between force and collected charge confirms the piezoelectric nature of the generated charge. **d** current profiles extracted from Fig. 3a for different applied forces. Applying 9 μN is not enough to read the current generated as it lies below the current threshold of the amplifier. As the force is increased, the amplifier responds to the generated charge

when the tip scans in the retrace direction a similar mechanism, with invert signs, occurs, see Fig. 1c. Spectroscopy experiments can also be performed, see Fig. 1d, as the tip exerting a force generates a positive charge (+Q), if an up domain is loaded, or a negative charge (−Q) if a down domain is loaded. By the contrary, the unloading process generates a negative charge (−Q) for an up domain and a positive charge (+Q) for a down domain. As current is being recorded, the rate at which the force is applied is crucial, as the current increases with force rate. Throughout the manuscript it is considered that a positive force (compressing force) will generate a positive current if applied into a positive (up) poled domain.

In the experiments we used a commercial probe with reference RMN-25PT200H. The tip is made out of a solid platinum wire consisting in an ultra-stiff cantilever, with spring constant of 250 N m$^{-1}$. Such fully metallic tip ensures that its conductivity is preserved while applying a high load and only a decrease in resolution can eventually occur. We tested the new mode on a typical reference material for PFM experiments which is a commercially available PPLN from Bruker AFM, in the form of a thick crystal Lithium Niobate with a reference bulk $d_{33}$ value of 7.5 pC N$^{-1}$. This material has been widely studied and its $d_{33}$ piezoelectric constant is reported to be in the range of 6–16 pC N$^{-1}$ for a Z-cut poled lithium niobate[46]. Additional ferroelectric materials were studied to test the DPFM method. A 400nm-thick BFO film prepared on a Platinum/Si substrate and with $d_{33}$ in the range of 16–60 pC N$^{-1}$ [47] was obtained from MTI Corp. A polycrystalline PZT with $d_{33}$ in the range of 57–97 pC N$^{-1}$ [48] was obtained from a commercial buzzer device. Before starting the measurements, the samples were scanned with the conductive tip

in order to discharge its surface from screening charges and minimize their effects[37, 49]. All the measurements were performed under an atmosphere of dry air (<8% humidity).

**Piezogenerated charge of periodically poled lithium niobate.**
Through the aforementioned set-up and the proposed physical explanation, we have been able to map piezoelectricity at the nanoscale. The output signal of the amplifier was both recorded at the Trace (Fig. 2a) and Retrace (Fig. 2b) scans. The images consist of a 256 × 128 pixels frames, 30 μm × 15 μm obtained at a speed of 0.01 lines per s (ln s$^{-1}$) (0.66 μm s$^{-1}$), including over-scan, recorded with a loading force of 234 μN. We used a particularly low speed to avoid scrapping surface screening charge which could interfere with the collected charge[36]. With these imaging parameters, the bandwidth needed to record current is 1–3 Hz, which is in accordance with what our amplifier can perform. The obtained images (Fig. 2a, b), show that the current is only recorded at the domain walls in accordance with the proposed physical model. A peak current of 15 fA is generated at the domain walls while its sign depends on the direction of the tip scan. We labeled the trace image, from left to right, as DPFM-Si, for Direct PFM Signal input, and the retrace image, from right to left, as DPFM-So, for Direct PFM Signal output. We have also tried to perform both PFM and DPFM methods, simultaneously. In order to do so, the back of the PPLN crystal was connected to the AC generator of the AFM, so an AC voltage signal was applied to the bottom surface of the sample maintaining a DC coupled ground. The PFM phase image is shown in Fig. 2c and PFM amplitude image is shown in Fig. 2d. The simultaneous

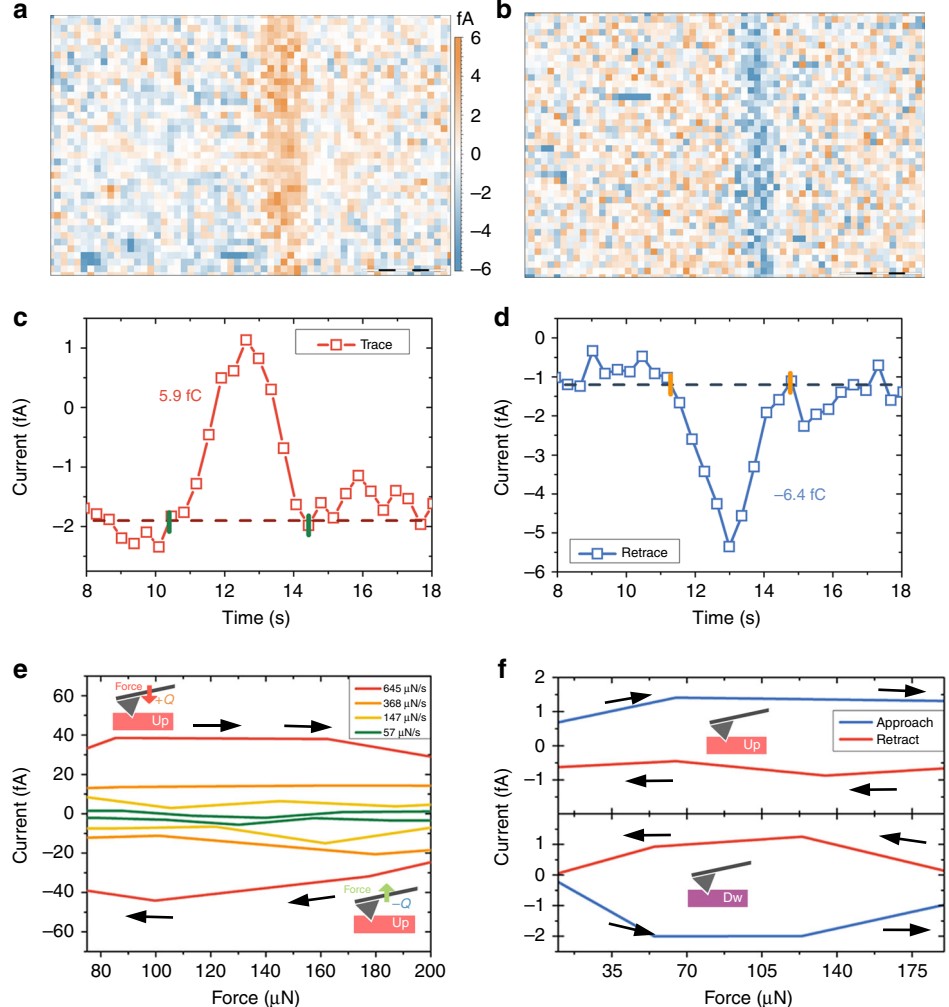

**Fig. 4** Quantitative measurements and spectroscopy experiments. **a** DPFM-Si and **b** DPFM-So images of a zoomed region of the PPLN sample recorded in order to fully integrate the charge generated, scale bar 1 μm. The mean profile average from the images was obtained in order to reduce noise. The resulting profiles are plotted in **c** and **d**, which are directly integrated to estimate the generated charge. **e** current-vs-force spectroscopy sweep performed in the Up domain configuration, where different force sweep rates where applied. The current generated increases with the increasing force rate. Its sign is the opposite for approach-when force increases- and retract-when force decreases. **f** current-vs-force spectroscopy sweep for an Up domain (top) and a Down domain (bottom). For an Up domain increasing the force will generate a positive current while the opposite occurs for a Down domain

acquisition of the four images of Fig. 2 shows how the DPFM mode can complement the standard PFM measurements providing, as we will discuss below, the data to quantify the piezoelectric coefficient of the material. Moreover, standard topography image obtained from contact mode operation is recorded (Supplementary Fig. 2). From DPFM-Si and DPFM-So images it is observed that there is a little gradient in the single domains areas, this will imply the collected current is not exactly zero. This could be due to different processes occurring simultaneously with piezoelectric charge generation, however its contribution is negligible compared to the peaks recorded at domains walls (Supplementary Fig. 3).

In order to obtain strong evidence of the piezoelectric origin of the current signal from the amplifier we prepared a full set of experiments related to the dynamics of piezoelectric charge generation. The charge generated from piezoelectric effect is known to be linear with the applied force[5]. This is a key aspect to distinguish piezoelectric charge from other possible charge generation phenomena[36, 50]. The relationship between current and applied load was tested by scanning the PPLN sample under

different applied loads, starting from a low loading force of 9 μN which was stepwise increased until reaching a maximum force of 234 μN. The recorded DPFM-Si and DPFM-So images are plotted in Fig. 3a, b, respectively. The tip speed was maintained constant along the whole image at a rate of 0.55 μm s⁻¹. We can observe that at the lowest load, no charge was collected by the amplifier, which was not capable of reading such a small current, i.e, between 0.1–0.3 fA for an applied force of 9 μN. The area recorded with the minimum force loading is also interesting to assess the influence of surface charge screening in the recorded currents. Before DPFM experiments, the sample was scanned with the same tip, at a tip speed 100 times faster, in order to fully discharge the sample surface from surface screening charge. The area scanned with 9 μN confirms that surface screening charges do not play an important role in the collected charge. If removal of surface screening charge through a scrapping process was important we should see a current in the 9 μN region, as the applied force is two-fold that needed to start the scrapping process[37]. Once the force is increased, the current recorded by the amplifier increases as well, as expected from a piezoelectric

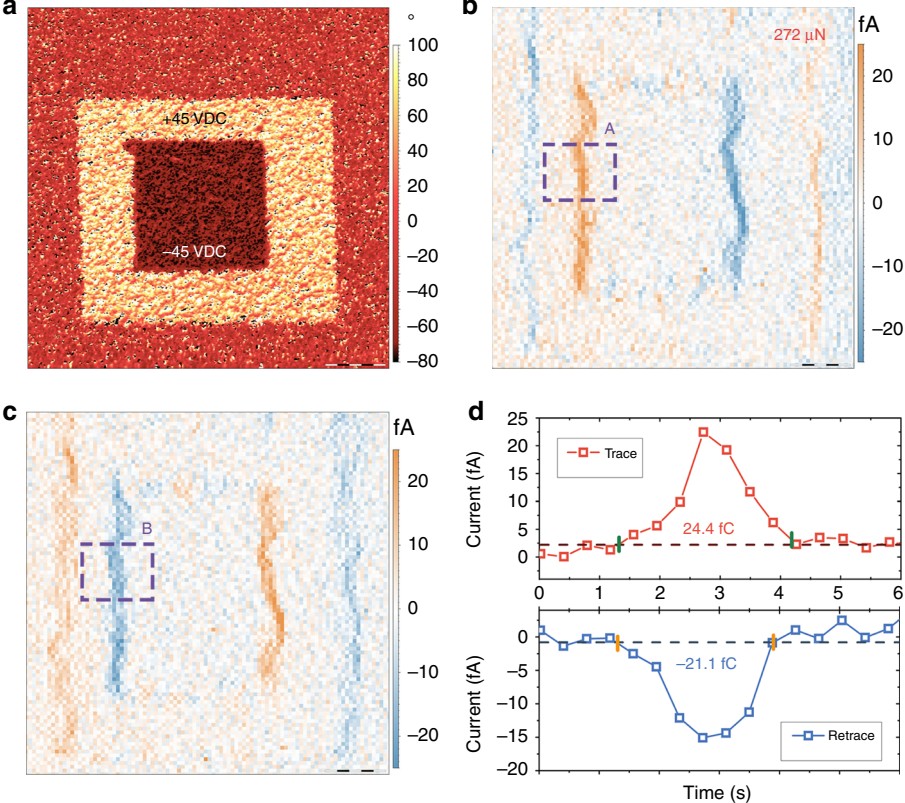

**Fig. 5** Piezogenerated charge of bismuth ferrite (BFO) **a** PFM phase image of a 400 nm thick $BiFeO_3$ (BFO) ferroelectric film in which two zones of opposite polarization were previously recorded. **b** DPFM-Si image and **c** DPFM-So image of the prerecorded areas, scale bar 2.5 μm. The inner, smallest square presents the largest current output, as we are crossing between two domains fully polarized in opposite directions. Accordingly, when crossing the large outer square border a lower current is recorded, as we are crossing from a virgin state to a fully polarized domain. **d** Mean-profile of the dot-line squared area A- in b- top and Mean profile of the dot-line square B of c. Both profiles where directly integrated to obtain the piezogenerated charge

generated charge. More importantly, the width of the current line generated at domain crossing does not substantially increase with applied load. The size of this line is not related to the domain wall thickness, but to a convolution effect caused by the tip[51, 52] (Supplementary Fig. 4).

In order to elucidate if the generated charge is proportional to the force we have analyzed the peak current values for DPFM-Si and DPFM-So frames, for each applied load. The maximum current values of a scan line were multiplied by the specific time constant of one pixel, which is 0.39 s, so the most of the piezoelectric charge is fully integrated. Finally, a relation between the collected Charge vs Applied load is found, which is plotted in Fig. 3c. A linear fit was used for both positive and negative charge, confirming the linear relationship between the generated charge and the applied force with Pearson's $R$ of 0.99 and −0.93 for each linear fitting. From the slope of this linear fit, an approximation of the $d_{33}$ piezoelectric constant of the material can be found with a value of 8.2 pC N$^{-1}$. The value obtained is an underrated approximation, as there is a part of the current generated that it is not being considered, as only the peak current is integrated. The current profile shape for each applied load was also analyzed, which are plotted in Fig. 3d. The profiles provide information on the dynamics of the charge generation at the nanoscale as the tip passes throughout the domain wall. The profiles, evidence that the increased generated charge for higher loads is related to the maximum current peak, rather than to the width of the Gaussian-like curve shape. Once the origin of the generated charge has been

proved to be the direct piezoelectric effect, we can now perform a mapping of the piezopower generation at the nanoscale with images of Fig. 2 (Supplementary Note 1 and Supplementary Fig. 5).

Obtaining quantitative values of piezoelectric and ferroelectric materials through an easy and reliable method is a high pursued target in the scientific community[53, 54]. In order to test if the method can be quantitative, we performed a zoomed-in image of a domain wall, recording both DPFM-Si and DPFM-So signals, see Fig. 4a, b. The images were performed with a tip speed of 0.22 μm s$^{-1}$ and an applied load of 234 μN. The zoomed-in images were sufficiently precise to fully integrate the generated current. In order to reduce thermal noise[55], the mean average profile for the total number of lines composing the image was obtained for both cases, see Fig. 4c, d. The resulting profile corresponds to the piezoelectric generated charge vs distance (μm), which, divided by the tip velocity, can be converted into charge vs time. With such experimental profiles, see Fig. 4c, d, we can perform a direct integration of the curves estimating the area beneath the curve and hence, the generated charge. We have found that the piezoelectric charge generated is 5.9 fC for DPFM-Si and -6.4 for the DPFM-So profiles. In order to see if the collected charge is a function of the tip speed we studied the evolution of the recorded charge vs. tip speed (Supplementary Fig. 6). The measured charge corresponds to a loading and unloading mechanism, and hence to find the piezoelectric charge we must divide this charge by a factor of two. The exact force exerted was calculated using a

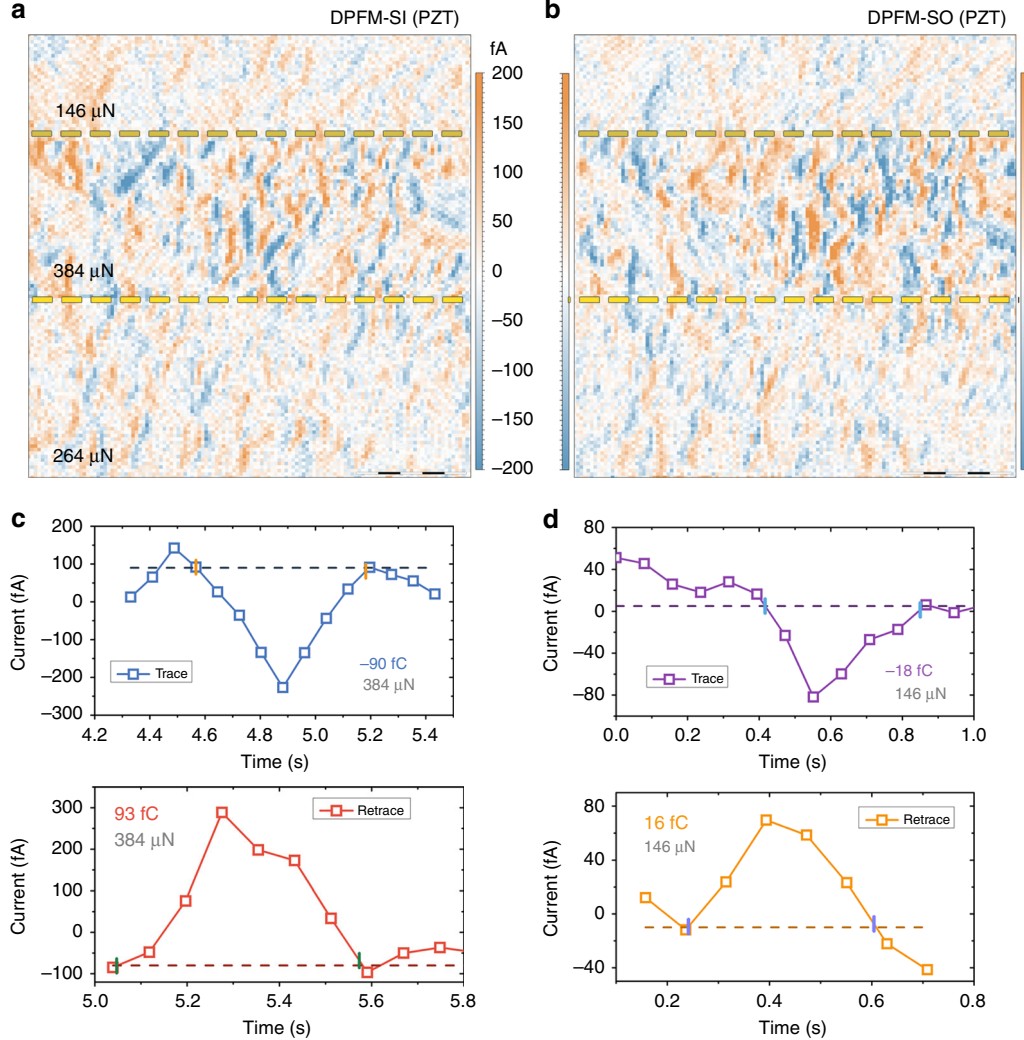

**Fig. 6** Piezogenerated charge of lead zirconate titanate (PZT) **a**, DPFM-SI and **b**, DPFM-SO of a PZT sample with natural domains, scale bar 2.5 μm. The force applied was increased at the middle of the scan, and then further decreased to see the current dependence upon the force applied. PZT was selected as it has a small surface charge density compared with BFO and PPLN but a higher $d_{33}$ constant. **c** and **d** are corresponding current profiles extracted from **a** and **b**. The $d_{33}$ constant, calculated by dividing the collected charge by the applied force is in agreement with values reported in the literature, and that the current recorded is larger compared with experiments performed on BFO and PPLN

force-vs-distance curve, (Supplementary Note 2 and Supplementary Fig. 7) and with such deflection sensitivity and the cantilever spring constant, the applied force was obtained. To diminish the error associated to the applied force, we have calculated the exact force constant of the probe used in the experiment, through a formula provided by the tip manufacturer and the real dimensions of the cantilever. Upon calculations, we found that the applied load is 234 μN, which yields a piezoelectric constant of $12.6\,\text{pC}\,\text{N}^{-1}$ and $13.6\,\text{pC}\,\text{N}^{-1}$, for DPFM-Si and DPFM-So, respectively. We evaluated the error that corresponds to the proposed method. The force error was found to be ± 9 μN, mainly caused by the determination of the spring constant of the cantilever. The charge measurement error was calculated as the sum of the noise spectral density error, the error created from the amplifier leakage current and the error obtained from the electrical calibration. We calculated the noise spectral density of our amplifier, which yields a value of 3 fA RMS. The standard error induced into the calculated current is found by averaging. Considering all the errors, we found that the $d_{33}$ piezoelectric constant of our sample is $13 \pm 3\,\text{pC}\,\text{N}^{-1}$ and $14 \pm 3\,\text{pC}\,\text{N}^{-1}$ for

DPFM-Si and DPFM-So respectively. As we are crossing the very same domain, we can use both quantities to acquire the final $d_{33}$ constant of the material as being $14 \pm 3\,\text{pC}\,\text{N}^{-1}$ standard error.

Spectroscopy experiments were performed to elucidate if the method could also be employed not only for imaging, but also as a tool of characterizing the piezoelectric response outside the ferroelectric domain walls or in non-ferroelectric piezoelectrics. For such purpose, the tip was placed in the middle of a ferroelectric domain and the current recorded while a force-vs-distance curve was obtained. The curve starts with a loading force of 5 μN and it is increased to a maximum value of 258 μN to go back to the initial 5 μN load. The current recorded from the amplifier was measured for different sweep rates, see Fig. 4e. As the force/time rate is increased, the recorded current increases as well confirming its direct relationship. Different spectroscopy events were obtained, see Fig. 4f; top which corresponds to a spectroscopy for up domain area and bottom for down domain area. It is found that for the up domain case, a loading curve will generate a positive current; however the current sign is the opposite in the case of a down polarization domain. The

spectroscopy curves start with the tip engaged under an applied load of 5 µN, to avoid collecting charge induced by electrostatic effects while the tip is placed into contact with the surface. For both curves a sweep rate of 53.2 µN s$^{-1}$ was employed. With such sweep rate, and by averaging the recorded current, we were able to estimate the $d_{33}$ coefficient (Supplementary Fig. 8). In order to clarify how the information is acquired, we further incorporated the force-vs-distance curves obtained within the Fig. 4e, f, (Supplementary Fig. 9). We have calculated the capacitive displacement current, using a parallel plate capacitor model, in the case of a full range spectroscopy curve (Supplementary Fig. 10 and Supplementary Note 3). The calculated current, was corroborated by performing spectroscopy curves in a non-piezoelectric sample (Supplementary Fig. 11). We further employed a piezoelectric PZT 5A1 in order to corroborate the piezoelectric origin of the current recorded (Supplementary Fig. 12) with the same spectroscopy conditions as for the non-piezoelectric sample.

**Piezogenerated charge of bismuth ferrite**. The feasibility of the method has been successfully demonstrated for a thick ferro-electric crystal with a low-intermediate piezoelectric $d_{33}$ constant. In order to check the performance of the DPFM method on other materials it was also tested on a 400 nm-thick BFO ferroelectric layer over platinum, commercially available from MTI Corp. The sample was previously scanned using PFM in order to record a pattern in its surface-the pattern is shown in PFM phase image of Fig. 5a, where DC voltages of +45 VDC and −45 VDC were applied to record the specific domain pattern. The same area was scanned using normal PFM mode in order to see if the domains can be read. Once recorded, DPFM-Si and DPFM-So images were performed, which are shown in Fig. 5b, c. It is found that the current generated appears only at the domain walls. However, at similar scanning parameters, it is found that the peak current is near 25 fA. BFO is a well-characterized ferroelectric material that has a Young modulus of 170 GPa and a surface screen charge of 80 µC cm$^{-2}$ [47]. These values are comparable to those of the previously tested PPLN[46]. However, the piezoelectric constant of BFO is significantly larger, between 16 and 60 pC N$^{-1}$ [47]. These differences in the measured $d_{33}$ constants can be used to explain the larger current that is recorded for BFO, as compared to PPLN. In order to discard imaging artifacts, the same pattern was reread in DPFM mode but rotating the scan direction, which rotates the image motives as well (Supplementary Fig. 13). It was found that the generated charge had its maximum value where the tip passes from a full polarized area to the opposite polarization direction. The capability of the mode to be quantitative was again tested by determining the $d_{33}$ value for the BFO sample. The same procedure as explained for Fig. 4c, d was employed for Fig. 5b, c. The squared areas in 5b, c were used to obtain an average of the lines composing squares resulting in the average profile of Fig. 5d. The top part corresponds to the A square and the bottom part corresponds to the B square. The values obtained are 24.4 and −21.1, which divided by the applied force, yield a $d_{33}$ values of $45 \pm 6$ pC N$^{-1}$ and $40 \pm 6$ pC N$^{-1}$ for DPFM-Si and DPFM-So profiles, which, averaged, result in a $d_{33}$ value of $43 \pm 6$ pC N$^{-1}$. Such value is in accordance with what is found in the literature, which ranges between 16 and 60 pC N$^{-1}$ [47], confirming the feasibility of the mode as a tool to quantify the piezoelectric coefficient. Spectroscopy experiments were also performed, with two different force rates applied (Supplementary Fig. 8).

**Piezogenerated charge of Lead Zirconate Titanate**. In order to discard any contribution from scraped charges as in CGM, we

studied as well a PZT sample obtained from a commercial buzzer device. The PZT sample was annealed at 150 °C, in a low humidity environment, to cross the Curie temperature and generate natural domains in the material[56, 57] (Supplementary Fig. 14). Then, we polished the PZT following standard procedures down to a thickness below 4 µm. We specifically selected PZT, as it is a well-known ferroelectric with a smaller surface charge density but a larger $d_{33}$ constant, compared to lithium niobate and BFO[58]. Hence, if DPFM would be mostly collecting the charge scraped from the surface, the integrated charges would be significantly smaller for PZT than for PPLN and BFO. In contrast, if the collected charge is originated by the direct piezoelectric effect, the integrated charges would be significantly larger for PZT than for PPLN and BFO. The DPFM-Si and DPFM-So images obtained for PZT are presented in Fig. 6a, b, where different forces were applied. We started the image, from the bottom, applying a force of 264 µN, up to 4 µm, where the force was increased to 384 µN. We then further decreased the force to 146 µN at 8 µm. From the data, we see that the current is much larger than those obtained for PPLN (Fig. 3) and BFO (Fig. 5). In order to quantify the collected charge for this set of measurements, we integrated several current profiles at different applied forces. The results, displayed in Fig. 6c, d, indicate that the charge collected is much larger than in the case of the BFO and PPLN for comparable forces. According to the CGM model, for PZT, one should expect a smaller collected charge resulting from surface charge scraping. However, this is not what is observed in our measurements, confirming that the collected charge in DPFM experiments is proportional to the piezoelectric coefficients of the tested materials. The average $d_{33}$ obtained from all the profiles is 88 pC N$^{-1}$, in accordance with the value reported in the bibliography[48]. More insight on the differences between CGM and DPFM can be found in the Supporting Information (Supplementary Note 4 and Supplementary Fig. 15), where we provide a comparison of different aspects of these two complementary modes.

## Methods

**Atomic Force Microscope**. The AFM equipment consists of a commercial unit, a Keysight 5500 LS. We employed the large area closed loop scanner with reference N9524A.

**Electronics**. The two operational amplifier were provided by Analog Devices INC, the transimpedance amplifier is populated with the following resistor MOX112523100AK, which is commercially available. The calibration procedure of the amplifier was performed using know resistor MOX-1125–23–4008J and the DC source from the AFM controller. The exact part number of the transimpedance amplifier is ADA4530–1 and the part number of the voltage amplifier is AD8429.

**Measurement conditions**. Low humidity was achieved both inside the AFM box and amplifier box, in order to reduce the leakage current present in the system. The tip employed for the measurements comprises a RMN-25PT200-H from Rocky Mountain Nanotechnology manufacturer. In order to record the BFO sample a homemade high voltage amplifier was used while the PFM image was obtained with RMN-25PT300 tip.

**Samples**. The PPLN sample was provided by Bruker INC, the BFO was provided by MTIXTL, PZT sample was provided by mayor electronic reseller and the PZT 5A1 was provided by Morgan Advanced Materials. All samples are commercially available.

**Data availability**. The data that support the findings of this study are available from the corresponding author on reasonable request.

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

## Acknowledgements

ICMAB acknowledges financial support from the Spanish Ministry of Economy and Competitiveness, through the "Severo Ochoa" Programme for Centres of Excellence in R&D (SEV- 2015–0496), and the projects Consolider NANOSELECT (CSD 2007–00041), MAT2014–51778-C2–1-R project, co-financed with FEDER, and the Generalitat de Catalunya (project 2014SGR213) as well as the French Agence Nationale pour la Recherche (ANR), projet Q-NOSS ANR ANR-16-CE09–0006–01. We thank ICMAB Scientific and Technical Services. We thank Oliver Anderson from Rocky Mountain Nanotechnology LLC for discussion on how to calibrate cantilever spring constant.

## Author contributions

A.G. carried out the measurements, analyzed the data, and wrote the results section of the manuscript with contribution from all co-authors. M.G. and A.C.G. wrote the

introduction section with contribution from all co-authors. A.G., M.G., A.C.G., T.P. and X.O. designed the experiments, discussed the results, and interpreted the data acquired.

## Additional information

**Competing interests:** The authors declare no competing financial interests.

