## [Peer Review File · Nature Communications]

Reviewers' comments:

Reviewer #1 (Remarks to the Author):

Overall, this is a very good piece of work on characterization method of piezoelectric materials based on AFM. The authors proposed a novel method using the direct piezoelectric effect to map the piezo-induced charge using AFM cantilever, which is totally different from the currently widely used PFM technique based on the converse piezoelectric effect. Furthermore, this DPFM mode can be operated together with the normal PFM mode, which could provide more nanoscale information on the domains and domain walls in ferroelectrics. I think most scholars in the PFM field will be interested with this work. Thus, I suggest its publication on Nat Comm. However, the following issues should be addressed before its acceptance.

1. The measurement of d_{33} on a single domain was studied using different pressing stresses in Fig.4(e), but the charge-stress curve were not provided from which the d_{33} can be determined.

2. The tip effect on the d_{33} measurement were not addressed at all. The stress field induced by the AFM tip is intensively inhomogeneous, which is similar to the case in nanoindentation. The wear problem of the AFM tip is much more severe than that of the nanoindentor, which makes the quantitative measurement rather difficult or impossible.

3. The authors did not provide the calibration process when measuring the d_{33} constant. The d_{33} of the PPLN can be measured more accurately by using a mature method. The range of 6-16pC/N is too large for a standard sample.

In summary, this work can be accepted after major revision.

Reviewer #2 (Remarks to the Author):

The authors claim the detection of piezoelectric charges created at nanoscale under the pressure produced by an SPM tip.

Recommendation: reject, as the current results do not sustain the claims of the authors.

General comments:

1.

there is no experimental novelty: similar experiments have been performed before (with a different equipment) [1, 2, 3] and similar results have been obtained, and on the same type of samples (ferroelectrics with domains). Therefore, these results are a mere confirmation of the previous experiments performed by the group at Argonne. The novelty inferred by the authors is the implementation of a more sensitive current amplifier, which allows to measure lower currents. While I consider this to be a significant achievement, in my opinion this is not a real scientific breakthrough.

2.

What appears a real novelty, is the authors' interpretation of data: the authors claim that the current measured while the AFM tip is crossing domain walls separating oppositely oriented ferroelectric domains is given by the charges generated via the direct piezoelectric effect by the pressure exerted by the AFM tip. In contrast, previous reports explain the same current by the surface charges collected/removed by the tip [1-3]. In my opinion the interpretation given by the authors is wrong, and the justification is as follows:

The contact between the AFM tip and the sample is typically approximated and modeled using the well-known problem in contact mechanics of a sphere pushed against a semi-infinite space (the Hertz model for contact dating back to 1882, see e.g. [4, 5]). The effective contact takes place in a circular, finite region, which can be calculated from the known parameters (radius of the sphere/tip, elastic properties of the two materials, and the contact force). This is the region from which the piezo-generated charges are collected. The stresses that appear in the contact region are always compressive and the same is valid when the indented material is piezoelectric, as calculated more than a decade ago. [6] Similarly, the stress is also compressive everywhere in the contact region for the more realistic case when a tangential force component is acting on the material (force arising due to the sliding friction force between tip and sample occurring during scanning) – see the textbooks [4-5].

Therefore, since the piezoelectric material is compressed EVERYWHERE in the contact circle, the charges generated and collected as the tip moves across the piezoelectric surface should have the same polarity (depending on the sign of the piezoelectric coefficient) everywhere in the contact region, and therefore should give a constant, non zero current during movement. This current should reverse the sign when the piezoelectric coefficient changes the sign (like for ferroelectric domains with opposite polarization orientation). The sign of the current should NOT depend on the scanning direction (trace/retrace).

However, this is not what the authors measure in their experiments: they measure NO current when scanning domains of either polarity. The explanation given by authors (lines 111-118 and presumably illustrated in Fig 1 a,b,c.) is wrong, because:

- there is NO "strained area to the right" and "unstrained area to the left", as justified above [4-5-6].

- even if there is an "unstrained area " on one side of the tip why are there charges generated there ??? if strain (or stress) is zero, then the piezo-generated charges are zero. Therefore, the charges generated in the "strained area" on the other side of the tip are NOT compensated (and thus a net current should be measured)

- the authors observe a peak of current only at domain walls, when the piezoelectric coefficient of the material under the tip changes its sign. This cannot be explained by the direct piezoelectric effect: If the measured current would be due to the piezoelectric charges, then the current should change the sign gradually going through zero.

Specific comments.

3.

Line 56-57: please note that PFM detection is not limited to the optical beam deflection system. The very first piezoresponse measurement was actually demonstrated using an STM [7].

4.

The authors provide plenty of technical details, but do not go deep into the science of the contact problem: for a contact force of 1 uN and a tip radius of 700 nm (estimated from Fig S4c) I calculate a contact radius of the order of 16nm and a maximum stress in that area of ~6GPa (with an average of 4GPa). The same stress is acting on the platinum tip, and therefore there is a problem: the Yield stress of platinum being ~0.32GPa, it cannot sustain such a stress. The authors should explain such inconsistencies.

5.

The only result that seems consistent with the piezoelectric scenario is Fig. 4 e and f, which shows a reversal of the current hysteresis upon pressing into domains of different polarity.

But here there is also a problem that was not considered: the same current could be given by the changing capacitance between tip/cantilever and sample: the surface potential of oppositely polarized domains having opposite signs [8] the displacement current induced by the varying capacitance (including the cantilever) with z would also show different signs in approaching/retracting curves. How do the curves in Fig 4e/f look when the tip is detached from

sample (above the contact zone)? What is the error of the data in those curves?

=====

- [1] Ref 44 (Hong et al, PNAS 2014)
- [2] Tong et al, ACS Nano 10, 2568–2574 (2016)
- [3] S Tong et al, Phys. Rev. Applied 3, 014003 (2015)
- [4] K.L. Johnson, Contact mechanics, Cambridge University Press.1985
- [5] V.L. Popov, Contact Mechanics and Friction. Physical Principles and Applications, Springer-Verlag, (2010)
- [6] E. Karapetian, M. Kachanov and S. V. Kalinin, Nanoelectromechanics of piezoelectric indentation and applications to scanning probe microscopies of ferroelectric materials, Philosophical Magazine 85, 1017 (2005)
- [7] H Birk et al J. Vac. Sci. Technol. B 9, 1162 (1991).
- [8] Kalinin et al, JAP 91, 3816 (2002)

Reviewer #3 (Remarks to the Author):

While the authors have presented a reasonably coherent story, and are reasonably excited about the ability to directly measure piezoelectric coefficients, I cannot recommend this paper for publication.

The authors have frankly suspiciously downplayed a nearly identical paper by Hong et al in PNAS. That paper is referenced, but only in regards to screening charges and a possible surface modification during scanning. In fact that work, and at least one similar one since then, presented almost the same results as those in this manuscript. This work moves the concept forward slightly with a seemingly better current sensor, and a more thorough analysis of the forces involved. But fundamentally, the difference is not sufficient to merit the authors' declaration of novelty, especially the 'first nanoscale images of the charge generated in a thick single crystal of Periodically Poled Lithium Niobate' as claimed in the abstract. In fact Hong et. al. made the same observation for PPLN.

Other comments:

The authors claim 'If removal of surface screening charge through a scrapping process was important we should see a current in the 9 μN region, as the applied force is two-fold that needed to start the scrapping process.' But this will depend strongly on the tip radius of curvature, tip modulus, and scan speed/gain settings. Is a 2-fold increase enough? How is the baseline for scrapping established?

What is meant by 'It is found that the piezoelectric current has a Gaussian-like shape, where the area below the Gaussian curve is the piezo-generated charge.' Is the current supposedly Gaussian vs time, or location? There is no particular reason for the current to be Gaussian vs. time, other than geometric convolution artifacts (i.e. actually vs location).

Structurally, the paper is written reasonably well, provided the grammar will be edited in several places.

"Scrapping" is written several times in the manuscript. The authors probably are referring to scrapping, which is a term somewhat uniquely used by Hong et al in their PNAS article.

Figure comments:

Figure 4: The authors should clarify why the current does not increase with force in e/f, but rather only with the approach/retract rate.

Figure 5: The authors should comment about why the basically straight edges of the written domain in (a) do not coincide with the curved edges in Figures b and c.

Figure S1: Two calibration curves are shown, b and c. What is the significance of presenting this data twice?

S2: Instead of a single friction image, it would be more meaningful to include lateral force microscopy images acquired in both trace and retrace directions. Or, the difference between the two.

More importantly, why is the friction data shown at all? The main text mentions that it is acquired, but there is no apparent reason for it. Also, the caption refers to a nonexistent Fig S2d.

S3: Recommend combining Plots b and c. It is redundant to show the current trace 3 times in the same figure.

Fig S4: The 'hertzian nanoindentation process' is mentioned in the caption. But why? The authors need to complete the argument about Hertzian indentation, presumably that the contact area is scaling with force. In that case, what is the predicted increase in contact area for the range of forces involved given the tip and sample moduli? Is the Hertzian model appropriate. It is probably more of a punch geometry in reality according to the post-SEM image in this figure.

Reviewer #1

(Remarks to the Author):

Overall, this is a very good piece of work on characterization method of piezoelectric materials based on AFM. The authors proposed a novel method using the direct piezoelectric effect to map the piezo-induced charge using AFM cantilever, which is totally different from the currently widely used PFM technique based on the converse piezoelectric effect. Furthermore, this DPFM mode can be operated together with the normal PFM mode, which could provide more nanoscale information on the domains and domain walls in ferroelectrics. I think most scholars in the PFM field will be interested with this work. Thus, I suggest its publication on Nat Comm. However, the following issues should be addressed before its acceptance.

We thank the reviewer for taking his/her time to review our manuscript. All the issues raised by the reviewer are addressed below

1. The measurement of d_{33} on a single domain was studied using different pressing stresses in Fig.4(e), but the charge-stress curve were not provided from which the d_{33} can be determined.

According to your suggestions, we have calculated the d_{33} constants from the spectroscopy curves which have been included in a new figure S8 of the revised manuscript. To do so, we calculated the average current generated, in fC/s , and divided by the specific force rate used, in $\mu N/s$, to obtain the d_{33} value. This procedure is equivalent to integrating all the current points and then dividing it by the specific applied force. The results presented in S8a are also given below:

Figure 1. Current vs Force spectroscopy sweep for an Up domain (top) and a Down domain (bottom) of the PPLN crystal. For an Up domain increasing the force will generate a positive current while the opposite occurs for a Down domain. Dot lines corresponds to the average current read by the amplifier, while the errors corresponds to statistical errors obtained.

We can now use the applied force rate, $53 \mu\text{N/s}$, to calculate the d_{33} , obtaining, for the four measurements presented in Figure 1, an average value of $15 \pm 6 \text{ pC/N}$. The error corresponds to the statistical errors and the additional errors of the electrical calibration and force calibration processes which were discussed in the manuscript. This d_{33} is, within the error bars, in agreement with that obtained from the imaging method. We have also included additional spectroscopy data in the supplementary information (S8b-c) regarding the BFO sample:

For the case of the BFO sample, the force rate is very similar, $60 \mu\text{N/s}$, however the current that is recorded is 3.7 ± 0.7 and -4.5 ± 0.8 fA. We calculated the d_{33} values dividing such current by the force rate which gave us 61 ± 14 and 75 ± 17 pC/N. Again, this is in agreement with the d_{33} the values obtained from DPFM images.

We would like to note that even though the spectroscopic method seems quantitative, provided that one makes use of long tips to avoid capacitive coupling and small ZPiezo travels to avoid electrostatic coupling, we want to be very cautious to draw such a conclusion. Further development of the technique is needed by performing more experiments to optimize the force rate to be used, and assess the validity of the spectroscopic method for quantitative determinations. Another figure was added to the two previous ones with even more spectroscopy data, see S8c.

2. The tip effect on the d_{33} measurement were not addressed at all. The stress field induced by the AFM tip is intensively inhomogeneous, which is similar to the case in nanoindentation. The wear problem of the AFM tip is much more severe than that of the nanoindenter, which makes the quantitative measurement rather difficult or impossible.

The reviewer is right in this observation. We have included in the manuscript, see supplementary S4, an assessment of the tip influence where we measure the tip shape after all the images were collected and from the

current maps, we also estimated the tip size for the high load applied. Definitely, we are applying a considerable load to the tip, and the tip, which is made of platinum, gets blunt. This reduces the lateral resolution of the mode. We selected these tips because they will preserve conductivity regardless whether the tip gets blunt or not. In contrast, metal coated tips could not be used for such purpose because the tip conductivity would vary as the tip is deformed. We have just purchased single crystal diamond boron doped tips, from the company Adama, which became recently available in the market. However, when we acquired new images we realized that these weren't anything better than our Pt tips from a tip-engineering point of view.

Regarding the effect on the piezoelectric constant, as it depends only on the force applied to the material and hence we did not expect any change into the specific piezoelectric coefficient values. What depends on the pressure and huge strain gradient are the mechanical properties of the sample itself. For instance, a high pressure can produce plastic deformation into the sample and affect the d_{33} value obtained. At this point, we did not see any major physical damage to the sample, according to the topography images obtained, at least with this particular kind of probes, so we dismissed possible mechanical plastic nanoindentation.

3. The authors did not provide the calibration process when measuring the d_{33} constant. The d_{33} of the PPLN can be measured more accurately by using a mature method. The range of 6-16 pC/N is too large for a standard sample.

We have used for these measurements a PPLN which is commercially sold as reference material for PFM measurements by Bruker (see <http://www.brukerafmprobes.com/a-3692-pfm-smpl.aspx>). The supplier reports the d_{33} of 7.5 pm/V i.e. 7.5 pC/N, for the bulk lithium niobate value. However, an accurate determination of d_{33} for PPLN is challenging using standard macroscopic techniques because the poling process results in an arrangement of up and down domains with opposite piezoelectric responses. In fact the possibility of overcoming this kind of limitations is one of the reasons why we developed DPFM. We indicated in the manuscript the details of this reference value from our PPLN supplier and we compared our determination of d_{33} with it.

In summary, this work can be accepted after major revision.

We thank the reviewer for considering our work for publication, hoping that we have answered all his/her requirements. In the revised manuscript we have included additional measurements of a PZT sample, in order to provide more conclusive evidence on the reliability of the proposed method.

Reviewer #2

General comments:

1.

there is no experimental novelty: similar experiments have been performed before (with a different equipment) [1, 2, 3] and similar results have been obtained, and on the same type of samples (ferroelectrics with domains). Therefore, these results are a mere confirmation of the previous experiments performed by the group at Argonne. The novelty inferred by the authors is the implementation of a more sensitive current amplifier, which allows to measure lower currents. While I consider this to be a significant achievement, in my opinion this is not a real scientific breakthrough.

We completely disagree with this view. The experiments performed in references 1,2 and 3 cited by the reviewer are completely different from the ones we performed. We scanned 1000 times more slowly and applied 100 times more force than in the aforementioned experiments. The equipment in references 1,2 and 3 of the reviewer's report cannot, by any means, measure femtoAmpere currents, and hence, they were unable extract the information that we have reported. In fact, the aim of these works was providing an alternative way of imaging ferroelectric domains which was faster than PFM. In contrast, the aim of our work was measuring the current related to the direct piezoelectric effect, opening the door to a nanoscale characterization of the piezoelectric coefficient. It is hard to understand how the reviewer can state that the results presented in our work are "a mere confirmation of previous experiments by the group at Argonne" as long as it is acknowledged in [1] that the contribution of piezoelectric charge in CGM experiments is negligible compared to the unscreened displacement charge, which is what is purposely collected by CGM mode. In our experiments we exclusively collected the piezoelectric charge which was not measureable in [1] and before starting the measurements, the samples were scanned with the conductive tip in order to discharge its surface from screening charges.

2.

What appears a real novelty, is the authors's interpretation of data: the authors claim that the current measured while the AFM tip is crossing domain walls separating oppositely oriented ferroelectric domains is given by the charges generated via the direct piezoelectric effect by the pressure exerted by the AFM tip. In contrast, previous reports explain the same current by the surface charges collected/removed by the tip [1-3].

This is a misleading statement. The reviewer should note that the currents in [1-3] are in the pA range while in our experiments the measured currents are in the fA range. Clearly these are not "the same current". In

fact, such difference of three orders of magnitude can be simply understood by considering a different origin of those currents: while the pA currents come from the collection of surface screening charges the fA currents that we measured arise from the stress exerted by the AFM tip through the direct piezoelectric effect. Again we would like to remind the reviewer that the existence of piezogenerated charges is indeed considered, although not measured, in [1] and calculated therein to be $\sim 10^{-2}$ fC for forces of $\sim 1 \mu\text{N}$ on PPLN. An analogous calculation considering the forces above $100 \mu\text{N}$ used in our experiments results, for PPLN, in piezoelectric charges in the order of fC, in agreement with our measurements.

In my opinion the interpretation given by the authors is wrong, and the justification is as follows:

The contact between the AFM tip and the sample is typically approximated and modeled using the well-known problem in contact mechanics of a sphere pushed against a semi-infinite space (the Hertz model for contact dating back to 1882, see e.g. [4, 5]). The effective contact takes place in a circular, finite region, which can be calculated from the known parameters (radius of the sphere/tip, elastic properties of the two materials, and the contact force). This is the region from which the piezo-generated charges are collected. The stresses that appear in the contact region are always compressive and the same is valid when the indented material is piezoelectric, as calculated more than a decade ago. [6] Similarly, the stress is also compressive everywhere in the contact region for the more realistic case when a tangential force component is acting on the material (force arising due to the sliding friction force between tip and sample occurring during scanning) – see the textbooks [4-5].

Therefore, since the piezoelectric material is compressed EVERYWHERE in the contact circle, the charges generated and collected as the tip moves across the piezoelectric surface should have the same polarity (depending on the sign of the piezoelectric coefficient) everywhere in the contact region, and therefore should give a constant, non zero current during movement. This current should reverse the sign when the piezoelectric coefficient changes the sign (like for ferroelectric domains with opposite polarization orientation). The sign of the current should NOT depend on the scanning direction (trace/retrace).

However, this is not what the authors measure in their experiments: they measure NO current when scanning domains of either polarity. The explanation given by authors (lines 111-118 and presumably illustrated in Fig 1 a,b,c.) is wrong, because:

- there is NO “ strained area to the right” and “unstrained area to the left”, as justified above [4-5-6].
- even if there is an “unstrained area “ on one side of the tip why are there charges generated there ??? if strain (or stress) is zero, then the piezo-generated charges are zero. Therefore, the charges generated in the “strained area” on the other side of the tip are NOT compensated (and thus a net current should be measured)
- the authors observe a peak of current only at domain walls, when the

piezoelectric coefficient of the material under the tip changes its sign. This cannot be explained by the direct piezoelectric effect: If the measured current would be due to the piezoelectric charges, then the current should change the sign gradually going through zero.

We disagree with the interpretation of the reviewer, but his/her comments have been useful to realize that Figure 1b-d and the corresponding explanation provided in the manuscript might be confusing and inaccurate and thus we have modified it. We thank the reviewer for his/her comments. In fact, there is some subtlety regarding the relationship between piezogenerated charges and the current recorded in our setup. It is important to note that, in contrast to what occurs in Charge Gradient Microscopy, piezogenerated charges are bound to the piezoelectric material. While the piezoelectric material is scanned with a given force the surface charge density generated by the tip is on average constant along the scan. In our setup the tip is grounded, and the amplifier only provides a current to keep the tip to ground voltage equal to zero, and this is the current we measure. Thus, a current is only collected when the tip encounters variations of surface charge density. In our manuscript such variations typically occur during increasing force ramps at a single point (Fig. 4) or at ferroelectric domain boundaries (Figs. 2,3,5,6). In the context of our work, which aims at showing the potential of the DPFM mode for the study of piezoelectrics, the details of the contact mechanics are not relevant, even though might be very relevant to define the limitations of the method in further studies.

2.

Line 56-57: please note that PFM detection is not limited to the optical beam deflection system. The very first piezoresponse measurement was actually demonstrated using an STM [7].

Please note that we did not state that PFM detection is limited by OBD System, but that OBD system is the most used system to perform PFM experiments.

3.

The authors provide plenty of technical details, but do not go deep into the science of the contact problem: for a contact force of 1 uN and a tip radius of 700 nm (estimated from Fig S4c) I calculate a contact radius of the order of 16nm and a maximum stress in that area of ~6GPa (with an average of 4GPa). The same stress is acting on the platinum tip, and therefore there is a problem: the Yield stress of platinum being ~0.32GPa, it cannot sustain such a stress. The authors should explain such inconsistencies.

The reviewer is right in this analysis of the fate of the tip but we do not consider this a major problem which can make our work inconsistent. Indeed, the tip gets blunt and we actually included a detailed explanation of this fact in the supporting information of the first version of the manuscript-see S4-. If the tip gets blunt we may lose lateral resolution,

which is a minor limitation of the technique, but it does not interfere in the piezoelectric coefficient quantification. This can be understood by considering the direct piezoelectric effect which consists in the generation of a dielectric polarization (which has dimensions of charge/area) by an applied stress (which has dimensions of force/area). Thus, the actual area is of little relevance in the determination of the collected charge dependence on the applied load.

4.

The only result that seems consistent with the piezoelectric scenario is Fig. 4 e and f, which shows a reversal of the current hysteresis upon pressing into domains of different polarity. But here there is also a problem that was not considered: the same current could be given by the changing capacitance between tip/cantilever and sample: the surface potential of oppositely polarized domains having opposite signs [8] the displacement current induced by the varying capacitance (including the cantilever) with z would also show different signs in approaching/retracting curves. How do the curves in Fig 4e/f look when the tip is detached from sample (above the contact zone)? What is the error of the data in those curves?

We were aware of the possible artifacts that could arise due to the changing capacitance of the tip cantilever and sample system and thus we used special tips with a large length (80 μm). Moreover, from our results, we can be confident that such artifact is not at play on two grounds:

- i) The piezoelectric coefficient have been determined by DPFM using two approaches, the constant force scans along ferroelectric domain boundaries and the increasing force ramps at a fixed point of the material. While the latter might be affected by the changing capacitance former is insensitive to this artifact. Pleasantly the d_{33} obtained from both approaches are in agreement, discarding the presence of a capacitive artifact in the force ramps determination of Fig. 4.
- ii) The d_{33} determination using force ramps was performed for PPLN and BTO. If a capacitive artifact was dominating these measurements one would expect a similar value for both materials as the capacitive signal variation depends on the surface potential, which should be of the same order of magnitude for PPLN and BTO as these materials have similar screening charges. In contrast the d_{33} coefficients that we obtained are different by almost one order of magnitude and in agreement with the values accepted in the literature.

Reviewer 3

While the authors have presented a reasonably coherent story, and are reasonably excited about the ability to directly measure piezoelectric coefficients, I cannot recommend this paper for publication. The authors have frankly suspiciously downplayed a nearly identical paper by Hong et al in PNAS. That paper is referenced, but only in regards to screening charges and a possible surface modification during scanning. In fact that work, and at least one similar one since then, presented almost the same results as those in this manuscript. This work moves the concept forward slightly with a seemingly better current sensor, and a more thorough analysis of the forces involved. But fundamentally, the difference is not sufficient to merit the authors's declaration of novelty, especially the "first nanoscale images of the charge generated in a thick single crystal of Periodically Poled Lithium Niobate" as claimed in the abstract. In fact Hong et. al. made the same observation for PPLN.

We thank the reviewer for the careful reading of our manuscript and for his/her sincerity regarding the opinion of our work.

Even though we disagree with the reviewer's judgement on that we would have "suspiciously downplayed a nearly identical paper by Hong et al. in PNAS.", we appreciate and welcome the reviewer's observations as an opportunity for better contextualising and highlighting the singularity and novelty of our work.

We acknowledge that our work and the PNAS article present some similarities but this is only because both make use of a Conductive AFM mode. However, both works are radically different in essence and in particular this divergence appears crystal clear upon making a detailed comparison of the aims, the main conclusions, the experimental setup and the experimental data of these works:

The aim of our work was demonstrating the possibility of using Conductive AFM techniques to perform quantitative measurements of the piezoelectric coefficients based on the direct piezoelectric effect, i.e. by measuring the current arising exclusively from the charges generated by this effect. In contrast, in the introduction of the PNAS paper one can read regarding the aim of that work: "we introduce charge gradient microscopy (CGM), a high-speed nanoscale tool to image ferroelectric and piezoelectric domains." (page 1, left column, lines 25-27), which was motivated by the need to find an alternative method to image ferroelectric domains as "one of the major drawbacks of PFM is that the speed of data acquisition is limited by the resonance frequency of the cantilever and the time constant of the lock-in amplifier" (page 1, left column, lines 19-22),

The main conclusion of our work is that with the approach and novel experimental setup we describe allow measuring the piezoelectric

coefficients through the direct piezoelectric effect using a conductive AFM tip. This implies a significantly smaller lateral resolution compared to other methods of characterization of piezoelectrics based on the direct effect. In the case of the PNAS paper, it concludes the following in its last paragraph: "In conclusion, CGM is a simple and fast scanning probe microscopy that can characterize polarization domains by scraping the screen charges on the surface using a conducting nanoscale tip. As such, CGM opens possibilities for investigating unscreened surfaces without the need of using ultrahigh vacuum systems. Furthermore, we envision that a CGM-based energy harvester can be designed."

Even more relevant to support our view is what can be concluded from a paragraph of the article in PNAS, acknowledging that the contribution of piezoelectric charge in CGM experiments is negligible compared to the unscreened displacement charge purposely collected in the CGM mode (lines 3 to 10 of the right column of page 2 of the PNAS work): "We also calculated the expected charge from direct piezoelectric effect, which is 0.007–0.019 fC under the load of 1.17 μ N for single-crystal Z-cut lithium niobate with a piezoelectric coefficient, d_{33} , of 6–16 pC/N (29). These calculations [45.4 fC (experiment) vs. 50.5 fC (unscreened displacement charge) and 0.019 fC (piezoelectric charge)] demonstrate that the measured signal is dominated by the unscreened displacement charge across domain walls at a scan frequency of 10 Hz". This implies that the current arising from the direct piezoelectric effect cannot be measured by the CGM approach.

Finally, in connection with the above comments, we would like to note the key differences between the current amplification setups of these two works, which explain that, in contrast to the PNAS work, we can reliably detect currents in the fA range. While Hong et al. make use of an inverter amplifier setup (see Fig.1 of PNAS article), our measurements rely on a setup involving an extremely sensitive transimpedance stage only recently available in the market (see Fig. 1a of our work). The deliberate choice of such an amplification design was made in accordance of the aim of our work and was crucial to the success of our measurements, which by no means could have been achieved with the setup of Hong et al.

In view of the reviewer's comments we realize that one can be easily lead to confusion due to the graphical similarity of Figs. 2a-c and 4a,b of the PNAS article and Fig. 2a,b of our work, in spite of the a difference of three orders of magnitude in the measured currents. In order to avoid misleading the reader, in the introduction of the revised version of our work we have cited the PNAS article of Hong et al. as a relevant example of study of ferroelectrics using conductive AFM and discussed the relevant differences between CGM and our DPFM method. Moreover, we have added new supplementary material comparing the expected results of both modes in different measuring conditions (see S10 in SI). Those differences are very significant and indicate that is possible to differentiate CGM and DPFM modes and that DPFM is a new mode on its own.

Regarding the criticism of reporting, as in the work of Hong et al., measurements on PPLN sample, the reviewer should consider that this material is a common reference ferroelectric material for the simplicity of its domain structure. In these regards we also studied, for their relevance, availability and characteristics other common widely-studied

ferroelectrics as BFO and, in the revised manuscript, PZT. In particular, the choice of PPLN and BFO was dictated by the fact that these materials present comparable surface screening charges and significantly different piezoelectric coefficients, which allowed us to be certain that the differences in the measurements corresponded to piezogenerated charges.

"first nanoscale images of the charge generated in a thick single crystal of Periodically Poled Lithium Niobate" is unfortunate because it does not specify the origin of the charge, but it was not our intention to downplay the work of Hong et al. In the revised version of the manuscript we have amended that sentence which now reads:

"We report the first nanoscale images of piezogenerated charge in a thick single crystal of Periodically Poled Lithium Niobate (PPLN), a Bismuth Ferrite (BiFO_3) thin film and Lead Zirconate Titanate (PZT)"

Other

comments:

The authors claim "If removal of surface screening charge through a scrapping process was important we should see a current in the 90° region, as the applied force is two-fold that needed to start the scrapping process." But this will depend strongly on the tip radius of curvature, tip modulus, and scan speed/gain settings. Is a 2-fold increase enough? How is the baseline for scrapping established?

We thank the reviewer for pointing this out. In fact the threshold was established for the CGM mode in all the CGM series of articles. In the first article, PNAS article by Hong et al., it is reported that all the surface charge is fully removed in a PPLN with 1 μN which is equivalent to 120 MPa. A further article on this mode (S. Tong et al. Phys. Rev. Appl. 3, 1 (2015)), investigates the pressure dependence of the scraped surface charge. The authors of these works actually measure peak currents and not a charge-as we do. However, following your advice, we have replotted our data and the data presented in that manuscript, which are presented below:

We can see from the above figure that in our measuring conditions we are far above the previously mentioned 120 MPa threshold. More importantly, the behaviour of both curves is completely different-mind that the scale is logarithmic scale.

In order to obtain the pressure in the graph, we divided the applied force by the contact area between the sample and the tip, assuming that the latter has a circular shape with a radius of 148.5 nm. This is the value calculated in supplementary information (see S4).

What is meant by ‘It is found that the piezoelectric current has a Gaussian-like shape, where the area below the Gaussian curve is the piezo-generated charge.’ Is the current supposedly Gaussian vs time, or location? There is no particular reason for the current to be Gaussian vs. time, other than geometric convolution artifacts (i.e. actually vs location).

Thank you for your comment. We fitted the data with Gaussian curve as it is a tip convolution effect-you are absolutely correct. The Gaussian like shape was used because, according to the Hertz deformation model, this is the shape that you should expect.

We also considered the integration of the whole current curve, directly, to calculate the area above the graphs. However, we think that the gauss amp fitting will give a more accurate value-even though that it hugely increases the calculated error of the measurements.

Structurally, the paper is written reasonably well, provided the grammar will be edited in several places. "Scrapping" is written several times in the manuscript. The authors probably are referring to scraping, which is a term somewhat uniquely used by Hong et al in their PNAS article.

Thank you for your comment, we have addressed these issues.

Figure comments:
Figure 4: The authors should clarify why the current does not increase with force in e/f, but rather only with the approach/retract rate.

Figures 4 e, f are measurements performed in scanning mode but at a single point of the material. In such conditions, with our setup, the only way to simultaneously apply a known force and record the piezogenerated charge is by applying force ramps while measuring the current. Since equal forces produce the same piezogenerated charges, the collected current will depend on the time interval to reach a given force level, and this will be larger for larger force rates, because the charge generated by this given force will be collected in shorter times.

Figure 5: The authors should comment about why the basically straight edges of the written domain in (a) do not coincide with the curved edges in Figures b and c.

This is an interesting observation. Fig. 5a is a PFM image that was performed 48h prior to the measurements of Fig. 5b using a different tip. Before performing the DPFM measurement a "discharge scan" of the material was performed with a force of 9 μN in order to remove screening charges. We believe that the observed roughening of the domain boundary in the DPFM image (Fig. 5b) might be related to a well known effect domain wall roughening due to the competition between elasticity and pinning under an applied stimulus driving the wall motion (see for instance P. Paruch and J. Guyonnet, *Comptes Rendus Physique* 14 (2013) 667). Most of the studies of these complex responses were focused on the effects of electric fields but strain gradients have been reported to a relevant aspect in the physics of domain wall roughening (see B. Ziegler et al. *Phys. Rev. Lett.* 111 (2013)247604) in the particular case of BFO thin films. Also BFO are known to present a particularly large roughness compared to other ferroelectric possibly due to its magnetoelectric coupling with magnetic domains (G. Catalan et al *Phys. Rev. Lett.* 100 (2008) 027602). In this context we hypothesize that the stress applied during the "discharge scan" might have the effect of increasing the roughness of the domain wall. While such effects are intriguing and very interesting, these are far beyond the scope of the present work.

Figure S1: Two calibration curves are shown, b and c. What is the significance of presenting this data twice?

The calibration curves obtained were acquired twice each in a different day, in order to be certain that this was reproducible. The calibration

used for the measurements was obtained from averaging the two calibrations.

S2: Instead of a single friction image, it would be more meaningful to include lateral force microscopy images acquired in both trace and retrace directions. Or, the difference between the two. More importantly, why is the friction data shown at all? The main text mentions that it is acquired, but there is no apparent reason for it. Also, the caption refers to a nonexistent Fig S2d.

The reviewer is right in that the friction image is not relevant. We have removed the mention to friction in the main manuscript and removed the image from the Supplementary Information. Unfortunately the lateral force was not recorded due to the limitation in the number of channels of the AFM. Indeed it would have been more reasonable to have acquired the lateral force instead of friction. The mention to Fig.S2d has been also removed.

S3: Recommend combining Plots b and c. It is redundant to show the current trace 3 times in the same figure.

This is indeed redundant and we have merged Figs S3b and S3c.

Fig S4: The ‘‘hertzian nanoindentation process’’ is mentioned in the caption. But why? The authors need to complete the argument about Hertzian indentation, presumably that the contact area is scaling with force. In that case, what is the predicted increase in contact area for the range of forces involved given the tip and sample moduli? Is the Hertzian model appropriate. It is probably more of a punch geometry in reality according to the post-SEM image in this figure.

We thank the reviewer for mentioning this. He have eliminated the mention to the hertzian model and now the sentence reads: ‘‘The increased thickness is partially due to tip degradation but also to the nanoindentation of Lithium Niobate by the tip.’’

Reviewers' comments:

Reviewer #1 (Remarks to the Author):

The authors had well addressed my comments (Reviewer #1). By reading their replies to the other two reviewers, I noticed that Reviewer #2 and #3 had misunderstand the novelty of this work. The authors' reply clarified all the issues and I think this work is truly valuable and should be published on Nat Comm.

A further minor suggestion to the authors: in subsequent works, the authors should measure the contact area of the tip before and after scanning. This can be done using the contact resonance principle in AFAM (atomic force acoustic microscopy). By measuring the contact resonance frequency on a known modulus sample, the contact area can be calibrated before scanning and the variations after scanning can be got.

Reviewer #2 (Remarks to the Author):

The authors revised their manuscript.

In my opinion, it may be published after the authors address the following points, particularly #4, #5 and #6 which are mandatory for a good quality future paper.

The comments' numbers (#1 - #4) below correspond to the numbers of my previous review.

#1. The experimental setup used is what is known as "conductive AFM", (at zero bias applied), composed of an AFM equipped with a conductive tip and an ampermeter (current amplifier, or transimpedance amplifier) and is therefore nothing new in the schematic diagram. In contrast to the authors, I do not consider that changing the experiment parameters (i.e. replacing the amplifier with a better one, scanning with different speeds and applying different forces) is a novelty.

Therefore, I disagree with the authors that scanning 1000 times slower while pressing 100 times harder is a "novel AFM based mode", or a "new technique".

(the reply on the statement that the screening charges were removed is below, as a new comment).

#2. The authors replied that the simple fact that the CURRENT measured is three orders of magnitude below that measured in charge removing experiments (CGM, see e.g. [1]) is a clear indication that the origin of the charge is different (i.e. of piezoelectric origin and not due to removing screening/surface charges).

I completely disagree: the current depends on the speed of the tip, and since in this manuscript the speed is 1000 times less than that in [1, 2] it is perfectly normal that the current is 1000 smaller when collecting the same amount of charge. The charge calculated here is of the order of 1-5 fC, and it has the same order of magnitude in [2] (i.e. around 10fC for passing across one domain wall). In the same paper it is shown that the peak current is linear with speed and the same conclusion can be drawn from Fig S6. Thus decreasing the speed 1000x is expected to result in a current of the order of few fA. Therefore, the current originating in the screening charges and that given by the piezoelectricity are comparable at low speeds. What makes them very different in [1, 2] is the much lower pressure/stress applied (reducing the piezoelectric contribution). The "subtlety" invoked by the authors is actually the very definition of the piezoelectric effect (apparition of a polarization =bound charge under stress)

I also disagree on the fact that the "details of contact mechanics are not relevant". It is exactly the very careful analysis of the strain distribution (and therefore induced D, electric displacement) in the contact region which made me realize that the current measured by the authors could indeed have piezoelectricity as origin. Therefore, I accept now that my original scenario (elaborated without these detailed calculations) was incorrect. However, there are some issues here (see below, as a new comment).

#2(duplicated#). The text in the manuscript reads "In this technique the sample vibration is determined by an optical beam deflection system, which is an indirect measurement". There is no indication that PFM could be performed using another detection method than the OBD. (the readership of Nature Communications is broad, not restricted to the ferroelectric community, for which PFM is familiar).

#3. OK

#4. The fact that the tip is longer (80 μm instead of the usual 10-20 μm) means a reduction in the cantilever-sample capacitance by maximum one order of magnitude. On another hand, the tip-sample capacitance might be much higher, as the tip is bulkier and less sharp than usual. I was really expecting to see force-distance AND current-distance curves (simultaneously acquired), for both approach/retract, AND for distances containing both contact and non-contact regions AND for both "up" and "down" domains (one single such force-distance curve – approach/retract - is already shown in Fig S7c, the sensitivity calibration). Only from these curves one can have an idea about the magnitude of the current given by changing the parasitic capacitance(s). The authors did not provide this data.

I am asking this because the data provided (current only, for a restricted range, Fig 4 e-f), doesn't look trustworthy: first, the curves appear composed of 4 (four) data points only. Second, the curves should be constant, but the dispersion of data appears larger than the mean value/error (particularly for the "down" domain). Does the low bandwidth of the current amplifier affect these fast measurements?

New comments

In view of my detailed analysis of contact mechanics involved (see #2), I agree now that the origin of the current measured could be piezoelectricity. However, I think the authors should address the following points in the manuscript:

#5

The main difference between the currents given by piezoelectricity and screening charges is their sign. (the screening charges have opposite sign to that of bound surface charges). This means that there is a simple way to establish which one dominates. That is, when the tip passes from a "down" domain into an "up" domain the peak current should be positive for the piezoelectric current and negative if the current comes from removing the screened charges. This indeed seems to be the case, according to Figs 1 and 2 of this manuscript, compared to Fig 4 of [2], but this aspect is not at all mentioned.

Therefore, the authors should clearly specify this difference in the manuscript.

However, there is an issue here: the authors should make sure that the assessment of the direction of polarization from the PFM phase is correct. This is because there is a significant difference between the PFM setup in this manuscript and that in [2]: in [2], the AC signal is applied to tip and a zero phase is associated to an "up" domain. In contrast, here the AC excitation is applied to bottom electrode, and they should measure a phase of 180deg for the same "up" domain. Instead, the phase reported in Fig 2 is -90deg. The authors should explain in more detail how they assessed the orientation of polarization, since this is the actual, real proof of the piezoelectric origin of the current.

Additionally, the gain of the amplifier is negative, i assume this was taken into account when assessing the sign of the current from its voltage output (it is not specified).

#6

The current measured is the sum of that given by piezoelectric charges and removal of screening charges. The screening charge density is equal (and opposite sign) to the normal component of electric displacement D_n (for complete screening). Very often, however, there is a partial screening only, with a coefficient $\alpha < 1$ [see, e.g. 3, also 1]. Furthermore, the charges relax in

time (after the surface charge is removed, it comes back from the ambient (for external screening, see the ample discussion in SI of [2]), since the unscreened surface is unstable. The relaxation time for charge screening depends on the availability of charges (/concentration) in the ambient atmosphere. For example, in their experiments, Kalinin et al. found a charge relaxation (redistribution) time of about 20 min, when spontaneous polarization was modified by changing the temperature. The same order of magnitude for the "re-screening" time (6-10 min) was found by Tong et al, for the case of the charges removed by scanning in contact mode with an AFM tip [1].

Since the authors do not specify anything about the environment of their experiment (in fact I cannot find the manufacturer of the AFM!), so I suppose it is the usual laboratory ambient atmosphere, thus I expect the surface screening charge to relax at the time scale of the experiment (0.01 Hz scan frequency would correspond to 30-40 min/image).

In order to support their interpretation, the authors state that prior to the current measurement, they removed the surface charge by scanning the surface (at high speed). To prove that the surface re-screening does not take place, the authors should provide data showing stable EFM or KPM contrast (as in [1] or [3]) after charge removal, at the same time scale, and in the same ambient conditions as the current measurement.

#7

page 2: the statement "to obtain a quantitative measurement of the piezoelectric constant in piezoelectrics" is misleading, since the method requires scanning over piezoelectric domains walls, which may be difficult to prepare/find (on non-ferroelectric materials such as quartz, ZnO, etc).

references:

1. Ref 38 (S Tong et al, Phys. Rev. Applied 3, 014003 (2015))
2. Ref 37 (Hong et al, PNAS 2014).
3. Kalinin et al. JAP 91, 3816 (2002).

Reviewers' comments:

Reviewer #1 (Remarks to the Author):

The authors had well addressed my comments (Reviewer #1). By reading their replies to the other two reviewers, I noticed that Reviewer #2 and #3 had misunderstand the novelty of this work. The authors' reply clarified all the issues and I think this work is truly valuable and should be published on Nat Comm.

A further minor suggestion to the authors: in subsequent works, the authors should measure the contact area of the tip before and after scanning. This can be done using the contact resonance principle in AFAM (atomic force acoustic microscopy). By measuring the contact resonance frequency on a known modulus sample, the contact area can be calibrated before scanning and the variations after scanning can be got.

We thank the reviewer for his/her comments, which we really appreciate.

Regarding the measurement of the contact area, we did not know the method based on contact resonance frequency; we will try to implement it in further papers.

Reviewer #2 (Remarks to the Author):

The authors revised their manuscript.

In my opinion, it may be published after the authors address the following points, particularly #4,#5 and #6 which are mandatory for a good quality future paper.

The comments' numbers (#1 - #4) below correspond to the numbers of my previous review.

We thank the reviewer for the time and reviewing our work. We have employed a huge effort in clarifying the concerns raised in the last review. We hope that with the new set of data, which is attached as separate answers to the questions, we can clarify the doubts of the reviewer about our findings.

#1. The experimental setup used is what is known as "conductive AFM", (at zero bias applied), composed of an AFM equipped with a conductive tip and an ampermeter (current amplifier, or transimpedance amplifier) and is therefore nothing new in the schematic diagram. In contrast to the authors, I do not consider that changing the experiment parameters (i.e. replacing the amplifier with a better one, scanning with different speeds and applying different forces) is a novelty.

Therefore, I disagree with the authors that scanning 1000 times slower while pressing 100 times harder is a "novel AFM based mode", or a "new technique".

(the reply on the statement that the screening charges were removed is below, as a new comment).

We claim a new AFM method as, following ISO 18115, the different AFM methods are classified according to the physical property measured rather than the experimental setup. Otherwise, Photoconductive AFM, Charge Gradient Microscopy, Current Sensing AFM, Constant amplitude STM or Soft Resiscope modes would have to be the same mode, as all of them are based in the use of a metallic tip attached to a current amplifier. As denoted in the literature these are considered different modes by the community of

scanning probe microscopies. Thus, we claim that the Direct Piezoelectric Force Microscopy is new mode as we are presenting the very first mapping of the direct piezoelectric effect with an AFM with nanometer scale resolution.

#2. The authors replied that the simple fact that the CURRENT measured is three orders of magnitude below that measured in charge removing experiments (CGM, see e.g. [1]) is a clear indication that the origin of the charge is different (i.e. of piezoelectric origin and not due to removing screening/surface charges).

I completely disagree: the current depends on the speed of the tip, and since in this manuscript the speed is 1000 times less than that in [1, 2] it is perfectly normal that the current is 1000 smaller when collecting the same amount of charge. The charge calculated here is of the order of 1-5 fC, and it has the same order of magnitude in [2] (i.e. around 10fC for passing across one domain wall). In the same paper it is shown that the peak current is linear with speed and the same conclusion can be drawn from Fig S6. Thus decreasing the speed 1000x is expected to result in a current of the order of few fA. Therefore, the current originating in the screening charges and that given by the piezoelectricity are comparable at low speeds. What makes them very different in [1, 2] is the much lower pressure/stress applied (reducing the piezoelectric contribution).

The “subtlety” invoked by the authors is actually the very definition of the piezoelectric effect (apparition of a polarization =bound charge under stress)

I also disagree on the fact that the “details of contact mechanics are not relevant”. It is exactly the very careful analysis of the strain distribution (and therefore induced D, electric displacement) in the contact region which made me realize that the current measured by the authors could indeed have piezoelectricity as origin. Therefore, I accept now that my original scenario (elaborated without these detailed calculations) was incorrect. However, there are some issues here (see below, as a new comment).

We thank the reviewer for this comment. The argument of a direct relationship between the current and tip speed holds true if the charge is maintained constant. However, references [1] and [2] given by the reviewer clearly show that the “scrapped” charge is not independent of the scanning speed, or is not the same for different scrapping tests . This is clearly illustrated by the original data of reference [1] cited by the reviewer, reported in S9 of the supplementary information, which is also presented below:

[Redacted]

If one integrates the current for the different curves, it is found that the charge is positive for those recorded at 5, 10 and 30 Hz, but negative for that recorded at 20 Hz. For the case of the 30 Hz curve, for instance, the first peak shows a positive charge, while for the second peak the integration reveals that the charge is almost zero. This clearly shows that one cannot generally assume that the screening charge is independent of the scanning speed, or that it is likely to present changes over space and time.

In the case of our data, several facts show that rather with screening charges we are dealing with bound charges arising from the direct piezoelectric effect:

- 1- In Figure 3, while no charge is collected in the area recorded 9 μN of force, the current increases along with the applied forces above 9 μN and, importantly, so does the charge because this image was recorded at a constant scanning rate. Since applying a higher pressure should increase the charge scrapping mechanism (see ref. 1 by Hong et al cited above), the most plausible explanation of the data in Figure 3 is that it is bound charge which originates from the direct piezoelectric effect and is indeed expected to increase with the applied strain. This is further confirmed by the following relevant fact
- 2- The tests performed in three different materials resulted in different amounts of collected charge: 5fC for PPLN, 25fC for the case of BFO and 90fC for the case of PZT. PZT is known to present less surface screening charge density compared to PPLN and BFO. In contrast its piezoelectric coefficient is larger than that of PPLN and BFO. This also indicates that the recorded charges stem from the piezoelectric effect and indeed the values of the piezoelectric coefficients calculated from our measurements are in good agreement with those reported in the literature.
- 3- Since the collection of surface screening charge depends upon the area of the tip, the higher the tip area, the higher the charge. We calculated that for the case of our tip, the screening charge that we should be collecting for PPLN would be 55fC if the screening charge was the predominant mechanism. In the area of 9 μN force we recorded virtually no charge, while the maximum recorded charge was 6.2fC, at the maximum applied force, which represents one order of magnitude difference from the theoretical surface screening charge mechanism.
- 4- The force dependence of the charge read by our amplifier (see Figure 3c) is linear as expected from the piezoelectric effect, while surface charge scrapping does not have such dependence. Moreover we note that as reported by Hong et al., surface charge scrapping mechanism saturated at 120MPa, while we recorded a progressive increment of the charge by increasing the applied pressures up to 3376MPa. This linear dependence between the force and the collected charge was not only demonstrated for PPLN but also for PZT (see Figure 6) with pressures that were above the 120 MPa threshold of the saturation of charge collection due to scraping mechanism.
- 5- Finally, with a progressive increase of the force at a fixed point (i.e spectroscopy experiment) we obtained piezoelectric coefficients in agreement with those measured in scanning experiments. In the revised manuscript we also included spectroscopy experiments for BFO (see Figure S8b,c). Note that screening charges cannot be sensed by spectroscopy experiments, and thus this effect is excluded.

#2(duplicated#). The text in the manuscript reads “In this technique the sample vibration is determined by an optical beam deflection system, which is an indirect measurement”. There is no indication that PFM could be performed using another detection method than the OBD. (the readership of Nature Communications is broad, not restricted to the ferroelectric community, for which PFM is familiar).

We corrected this sentence to make it broader for the readership of Nature Communications.

#3. OK

#4. The fact that the tip is longer (80 μm instead of the usual 10-20 μm) means a reduction in the cantilever-sample capacitance by maximum one order of magnitude. On another hand, the tip-sample capacitance might be much higher, as the tip is bulkier and less sharp than usual. I was really expecting to see force-distance AND current-distance curves (simultaneously acquired), for both approach/retract, AND for distances containing both contact and non-contact regions AND for both “up” and “down” domains (one single such force-distance curve – approach/retract - is already shown in Fig S7c, the sensitivity calibration). Only from these curves one can have an idea about the magnitude of the current given by changing the parasitic capacitance(s).

The authors did not provide this data.

I am asking this because the data provided (current only, for a restricted range, Fig 4 e-f), doesn't look trustworthy: first, the curves appear composed of 4 (four) data points only. Second, the curves should be constant, but the dispersion of data appears larger than the mean value/error (particularly for the “down” domain). Does the low bandwidth of the current amplifier affect these fast measurements?

Regarding the displacement currents of the cantilever we provided an estimation of such effect in the supplementary info, see new Figure S13. In order to calculate such current level, we employed the most common model in AFM, the parallel capacitor modeling -for instance, such is the model used in Kelvin Probe Force Microscopy. We reproduced the new supplementary info here:

“

S10

In order to confirm that the current collected in spectroscopy measurements is related to the piezoelectricity of the sample and to exclude artifacts from sample-cantilever capacitive coupling, we performed a calculation of the displacement currents for the cantilever. With such calculation, we want to obtain the current due to the changes of the cantilever-sample capacitance during spectroscopy curves. We modeled the cantilever and sample as a parallel plate capacitor, as shown in figure S10.

Figure S10. Scheme of the Cantilever and sample setup used to calculate the tip-sample capacitance as a parallel plate capacitor.

We used the following parameters, $L = 200 \mu\text{m}$, $b = 35 \mu\text{m}$, $h = 80 \mu\text{m}$ and dielectric constant, $\epsilon = 8,8 \cdot 10^{-12} \text{ F/m}$. If we apply the known capacitance expression for a parallel plate capacitor, we can find:

$$C = \epsilon \frac{A}{h}$$

Where A is the area of the capacitor, in our case

$$A = L * b$$

Performing the calculation, we obtain that the capacitance is $7,7 \cdot 10^{-16} \text{ F}$.

The previous calculations correspond to the initial state of the lever. We now calculate the capacitance at the end of the spectroscopy curve, by assuming that the whole cantilever approaches to the sample surface by $5 \mu\text{m}$, so the new parameters are $L = 200 \mu\text{m}$, $b = 35 \mu\text{m}$, $h = 75 \mu\text{m}$, $\epsilon = 8,8 \cdot 10^{-12} \text{ F/m}$.

With such parameters, the capacitance is now $8,2 \cdot 10^{-16} \text{ F}$.

The capacitance variation is now:

$$\Delta C = 5,13 \cdot 10^{-17} \text{ F}$$

The voltage difference between the capacitor plates stores a charge in the capacitor. We use here a value of 3 Volts, an approximation of the work function difference between the plates. In reality we are always in contact with the material, so such 3V value is an overestimation of the voltage difference. With such parameters, we can calculate the charge as:

$$\Delta Q = \Delta C * V = 1,54 \cdot 10^{-16} \text{ C.}$$

Such charge, if measured in 3 seconds (the time employed for curves of Figure 4f), will produce a current of 0,05 fA which is almost two orders of magnitude smaller than the experimentally recorded currents.

“

Regarding the lack of points in the spectroscopy curves, as the referee suggests, the bandwidth and thermal noise of the amplifier is the limiting factor here. For the spectroscopy curves, we equipped the amplifier with a new feedback resistor which boosts the bandwidth of the system-considering a new calibration procedure for the amplifier. With this new configuration, we decided to test the calculation of the displacement current, by analyzing a NON-piezoelectric sample, as a microscopy glass slide. Here is the data that we incorporated in the supplementary info:

“**S11**

In order to validate the calculation of S10, we performed spectroscopy measurements in a non-piezoelectric sample, a glass microscopy slide. In order to increase the current induced by capacitive coupling, we used the extreme case where the Z piezo range is 10 μm . This experiment was carried out For this case we populated the transimpedance amplifier with a 10GOhm resistor which boosts the bandwidth of the overall system to 159 Hz. Such change also increases the current noise of the amplifier in one order of magnitude as well as increasing the 50Hz noise pickup. The data is presented in **Figure S11a**, while in **Figure S11b** we show an average of all the curves with its standard deviation, see. From our data, it is not possible to distinguish which are the approach or the retract curves, confirming that the displacement current due to the changes in the sample-cantilever capacitive coupling cannot be measured by the amplifier.

.”

Figure S11a, Current (fA) vs Time (s) obtained for a glass slide test sample and **Figure S11b** Current (fA) vs Time (s) where we average all the curves. There is no difference between the approach and the retract curves for such series of measurements confirming that the displacement current is not measurable by the amplifier.

“

At this point, we decided to try this configuration with a piezoelectric sample, and we performed spectroscopy experiments in such samples. The results are also included in new figure S15 of supplementary material, which we also reproduce here:

“

S12

The measurements performed in S11 correspond to a non-piezoelectric sample, so we need to test the system with a known piezoelectric sample in order to corroborate that the electronics are correctly working. In order to do so, we specifically prepared two samples, comprising a PZT 5A1 from Morgan Advanced Materials, a lead zirconium titanate (PZT) with top and bottom gold electrodes. The PZT sample was broken in half, and one of the halves was turned upside down in order to obtain an “up” polarized sample and a “down” polarized sample. Then both pieces were polished, with standard polishing procedure,

removing the top electrode from the ceramic layer. Then we performed an analogous experiment as in S11 for both pieces of PZT 5A1. We can see, Figure S12a and S12b, that both approach and retract curves created their corresponding current profiles, considering that each curve was performed in different points of the sample surface. Note that, as expected from the direct piezoelectric effect, the current profiles of the approach and retract curves corresponding to opposite dielectric polarizations present inverted signs. In order to diminish the errors from the curves, we averaged each of the curves, data is presented in Figure S12d and S12e.

Figure S12a scheme showing the preparation of the test samples. The poled piezoelectric sample was broken in half, one of the halves was flipped 180° and both top electrodes were removed by polishing in order to uncover the ceramic layer underneath. **S12b** Current (fA) vs Time (s) curves performed in a full Approach and Retract cycle, where the approach cycle is depicted in red color, while the blue one corresponds to the retract cycle. Figure **S12c** Current (fA) vs Time (s) which corresponds to the down polarized sample. **S12d** Current (fA) vs Time (s) for the average over multiple curves obtained for S12b and **S12e** Current (fA) vs Time (s) by averaging curves of S12c.

The reviewer also asked for the original Force-vs-Distance curves, which we have included in the supplementary info, see S12:

“**S9**”

The spectroscopy experiments were carried out by recording normal Force-vs-Distance curves, while at the same time, we recorded the ADC channel to acquire the signal from the amplifier. In order to make clearer the reproducibility of such curves, we incorporated in this supplementary information all the Force-vs-Distance curves that correspond to the Curves of Figure 4e and 4f of main text. We did not include the current channel, as it is included in the figures at the main text. Within this data we can see the reproducibility of the curves along each of the different force sweep rate applied.

Figure S9, Deflection (V) vs Distance (μm) for each of the spectroscopy experiments included in the manuscript. **S9a** corresponds to the curve of Figure 4f UP domain, **S9b** corresponds to the curve of figure 4f DOWN domain, **S9c** corresponds to the curve of figure

4e at 645 $\mu\text{N/s}$, **S9d** corresponds to the curve of figure 4e at 368 $\mu\text{N/s}$, **S9e** corresponds to the curve of figure 4e at 147 $\mu\text{N/s}$ and **S9f** corresponds to the curve of figure 4e at 57 $\mu\text{N/s}$.

“

We have also created the graphs, Force and Current –vs-Distance curves for Figure 4f, which are presented below:

The reviewer also required curves containing touching and non-touching areas of the sample surface. We carried out such experiments as well, here are the results:

Unfortunately, following the reviewer suggestions we can no longer have repetitive spectroscopy curves, each of them is different, there is no tendency. In the image, each of the yellow squares represents where we performed a single Force-vs-Distance curve. So each of such curves were performed in different spots of the sample surface. This is the same procedure performed for the glass and PZT 5A1. The first channel is the force, while the second channel is the output from the amplifier. We designed our measuring procedure so that the tip starts touching the sample, ensuring that the electronic amplifier has a ground and thus stabilizing internal capacitance of the amplifier. Otherwise, the instrument would not have a ground and hence the current recorded would be composed of residual noise pickup. In order to obtain repetitive data the tip needs to be in contact with the material all the time, and from that point, the force is increased or reduced while the current is recorded at the same time.

New comments

In view of my detailed analysis of contact mechanics involved (see #2), I agree now that the origin of the current measured could be piezoelectricity. However, I think the authors should address the following points in the manuscript:

#5

The main difference between the currents given by piezoelectricity and screening charges is their sign. (the screening charges have opposite sign to that of bound surface charges). This means that there is a simple way to establish which one dominates. That is, when the tip passes from a “down” domain into an “up” domain the peak current should be positive for the piezoelectric current and negative if the current comes from removing the screened charges. This indeed

seems to be the case, according to Figs 1 and 2 of this manuscript, compared to Fig 4 of [2], but this aspect is not at all mentioned.

Therefore, the authors should clearly specify this difference in the manuscript.

However, there is an issue here: the authors should make sure that the assessment of the direction of polarization from the PFM phase is correct. This is because there is a significant difference between the PFM setup in this manuscript and that in [2]: in [2], the AC signal is applied to tip and a zero phase is associated to an “up” domain. In contrast, here the AC excitation is applied to bottom electrode, and they should measure a phase of 180deg for the same “up” domain. Instead, the phase reported in Fig 2 is -90deg. The authors should explain in more detail how they assessed the orientation of polarization, since this is the actual, real proof of the piezoelectric origin of the current.

Additionally, the gain of the amplifier is negative, I assume this was taken into account when assessing the sign of the current from its voltage output (it is not specified).

We thank the reviewer for these comments. Indeed, we actually had the same doubts during the measurements as the PFM phase can be misleading depending on where you apply the AC bias. That is why we included the BFO sample, where we specifically wrote the domain pattern that we wanted. Such domain pattern was specifically written so that we were sure which were the up and down domains. We have made it clearer in the text. We want to note that according ref. [2] cited by the reviewer, the charge, not the current, is no longer positive for all the charge scrapping mechanisms as denoted by all the proposed physical mechanisms to explain the different recorded positive and negative charges.

Regarding the gain of the amplifier, we included a supporting information section specifically devoted to the calibration of the amplifier. You can see that in figure S1b we took into account the inverting gain of the current amplifier-and both the inverting gains of each of the inverting voltage amplifier.

#6

The current measured is the sum of that given by piezoelectric charges and removal of screening charges. The screening charge density is equal (and opposite sign) to the normal component of electric displacement D_n (for complete screening). Very often, however, there is a partial screening only, with a coefficient $\alpha < 1$ [see, e.g. 3, also 1]. Furthermore, the charges relax in time (after the surface charge is removed, it comes back from the ambient (for external screening, see the ample discussion in SI of [2]), since the unscreened surface is unstable. The relaxation time for charge screening depends on the availability of charges (/concentration) in the ambient atmosphere. For example, in their experiments, Kalinin et al. found a charge relaxation (redistribution) time of about 20 min, when spontaneous polarization was modified by changing the temperature. The same order of magnitude for the “re-screening” time (6-10 min) was found by Tong et al, for the case of the charges removed by scanning in contact mode with an AFM tip [1].

Since the authors do not specify anything about the environment of their experiment (in fact I cannot find the manufacturer of the AFM!), so I suppose it is the usual laboratory ambient atmosphere, thus I expect the surface screening charge to relax at the time scale of the experiment (0.01 Hz scan frequency would correspond to 30-40 min/image).

In order to support their interpretation, the authors state that prior to the current measurement, they removed the surface charge by scanning the surface (at high speed). To prove that the surface re-screening does not take place, the authors should provide data showing stable EFM

or KPM contrast (as in [1] or [3]) after charge removal, at the same time scale, and in the same ambient conditions as the current measurement.

We thank the reviewer for these comments. In the manuscript, we included an experimental section which describes how we performed the measurements but the reviewer is right to point out that this does not include relevant data such as the AFM model and the atmosphere under which we performed the measurements. We apologize for not giving these details which have been included in the second revision of the manuscript. This the new specific chapter summarizing all the experimental conditions is copied below:

“

EXPERIMENTAL CONDITIONS

The AFM equipment consists of a commercial unit, a Keysight 5500 LS. The two operational amplifier were provided by Analog Devices INC, the transimpedance amplifier is populated with the following resistor MOX112523100AK, which is commercially available. The calibration procedure of the amplifier was performed using know resistor MOX-1125-23-4008J and the DC source from the AFM controller. Low humidity was achieved both inside the AFM box and amplifier box, in order to reduce the leakage current present in the system. The tip employed for the measurements comprises a RMN-25PT200-H from RockyMountain Nanotechnology manufacturer. The PPLN sample was provided by Bruker INC, the BFO was provided by MTIXTL, PZT sample was provided by mayor electronic reseller and the PZT 5A1 was provided by Morgan Advanced Materials. In order to record the BFO sample a homemade high voltage amplifier was used while the PFM image was obtained with RMN-25PT300 tip.

“

Regarding the screening recharging time, we employed low humidity ambient conditions in the AFM. According to AIP Advances 6, 015220 (2016) the screening time removal occurs in hours, not in minutes. The difference between our setup and that of CGM is that we used low humidity ambient conditions (dry air) under which it has been shown that the rescreening process hugely decelerates (JOURNAL OF APPLIED PHYSICS 113, 187213 (2013)). The measuring conditions, in essence, are opposed to the CGM, basically, we work with low humidity ambient conditions for both reducing the leakage current of the amplifier and reducing the screening recharge mechanism.

Such surface recharge effect which according to the aforementioned reference occurs in the timescale of hours, is higher than our acquisition time. If surface screening charges would have a major contribution in our measurements we could not get the data reported-see the S11 of supplementary information.

#7

page 2: the statement “to obtain a quantitative measurement of the piezoelectric constant in piezoelectrics” is misleading, since the method requires scanning over piezoelectric domains walls, which may be difficult to prepare/find (on non-ferroelectric materials such as quartz, ZnO, etc).

We corrected the sentence circumscribing it to ferroelectric materials. We think that the spectroscopy, with the correct experimental conditions, can be quantitative as well but a significant amount of new work would be needed, which is outside the scope of this article.

references:

1. Ref 38 (S Tong et al, Phys. Rev. Applied **3**, 014003 (2015))
2. Ref 37 (Hong et al, PNAS 2014).
3. Kalinin et al. JAP **91**, 3816 (2002).

REVIEWERS' COMMENTS:

Reviewer #2 (Remarks to the Author):

the authors did their best to answer my previous concerns.

After a very careful judgement of their replies, including additional explanations and data, I find the new form of the manuscript suitable for publication.